# GNSS RUMS: GNSS Realistic Urban Multiagent Simulator for Collaborative Positioning Research

**Guohao Zhang** [1] [ID], **Bing Xu** [1] [ID], **Hoi-Fung Ng** [1] **and Li-Ta Hsu** [1,2,*] [ID]

1   Interdisciplinary Division of Aeronautical and Aviation Engineering, The Hong Kong Polytechnic University, Kowloon, Hong Kong, China; guo-hao.zhang@connect.polyu.hk (G.Z.); pbing.xu@polyu.edu.hk (B.X.); ivannhf.ng@connect.polyu.hk (H.-F.N.)
2   Research Institute for Sustainable Urban Development, The Hong Kong Polytechnic University, Kowloon, Hong Kong, China
*   Correspondence: lt.hsu@polyu.edu.hk

**Abstract:** Accurate localization of road agents (GNSS receivers) is the basis of intelligent transportation systems, which is still difficult to achieve for GNSS positioning in urban areas due to the signal interferences from buildings. Various collaborative positioning techniques were recently developed to improve the positioning performance by the aid from neighboring agents. However, it is still challenging to study their performances comprehensively. The GNSS measurement error behavior is complicated in urban areas and unable to be represented by naive models. On the other hand, real experiments requiring numbers of devices are difficult to conduct, especially for a large-scale test. Therefore, a GNSS realistic urban measurement simulator is developed to provide measurements for collaborative positioning studies. The proposed simulator employs a ray-tracing technique searching for all possible interferences in the urban area. Then, it categorizes them into direct, reflected, diffracted, and multipath signal to simulate the pseudorange, $C/N_0$, and Doppler shift measurements correspondingly. The performance of the proposed simulator is validated through real experimental comparisons with different scenarios based on commercial-grade receivers. The proposed simulator is also applied with different positioning algorithms, which verifies it is sophisticated enough for the collaborative positioning studies in the urban area.

**Keywords:** global navigation satellite system (GNSS); simulator; collaborative positioning; multipath; 3D building model; urban canyon



## 1. Introduction

The development of intelligent transportation is one of the essential objectives of building a smart city, namely smart mobility [1]. The intelligent transportation system (ITS) aims to alleviate traffic jams and enhance traffic efficiency with sufficient assistance and planning. However, as the foundation of ITS, the precise navigation of transportation participants can hardly be achieved, especially in the dense urban area [2]. Most of the navigation algorithms rely on the global navigation satellite system (GNSS) for functionality or initialization, since it is the primary sensor directly providing the absolute positioning solution. Besides the systematic interferences that can be adequately modeled (e.g., atmospheric delay and ephemeris error), the GNSS measurement can be degraded by other effects related to the environment, resulting in an enormous positioning error unqualified for ITS [2,3]. On the other hand, the GNSS precise point positioning (PPP) [4,5] or real-time kinematic (RTK) [6] technique employing the carrier phase measurement can achieve submeter level accuracy. However, the visibility limitation and cycle-slip phenomenon in the urban area will significantly degrade their reliability [7,8]. Although various approaches, such as the integration of GNSS with INS, camera [9], or LiDAR-based simultaneous localization and mapping (SLAM) [10,11], are capable of achieving better positioning accuracy, they still require an accurate absolute positioning solution from GNSS

for initialization. It is inevitable to improve the GNSS positioning accuracy in the urban area prior to ITS development.

Due to the rapid development of vehicle-to-everything (V2X) communication technology [12], the collaboration of transportation participants to achieve accurate positioning solution becomes possible, namely the collaborative positioning (CP). In most CP techniques, the neighboring road agents (GNSS receivers) measure and share the interagent ranges as well as their positioning solutions, in order to optimize the overall positioning solution of each involving agents. By making use of the additional interagent information, the CP can achieve higher positioning accuracy compared with conventional standalone solution [13–15]. Another straight-forward benefit of employing CP is to reduce the common noise by involving more road agents during positioning [16]. Moreover, CP can make use of the network geometry formed by road agents to improve the overall positioning optimization [17]. Unlike the transponder-based CP approach using DSRC or UWB, the GNSS-based CP can maintain the collaboration even when there exist obstacles between agents [18]. Among various GNSS-based CP methods, a popular approach is to apply double difference (DD) on the pseudorange measurements from neighboring road agents to eliminate the systematic error during relative positioning, such as atmospheric delay and ephemeris error [19,20]. Besides, based on the angular geometry between satellites and agents, the interagent range can be estimated for CP even when some of the satellite signals are blocked by buildings [21]. However, these methods can be significantly degraded by the multipath and NLOS reception error. Recently, the 3D mapping aided (3DMA) GNSS technique has been developed, using the 3D building model to predict and mitigate those errors in the urban area. Two representative approaches are the 3DMA GNSS shadow matching based on the satellite visibility prediction [22] and the 3DMA GNSS ray-tracing based on the LOS/NLOS pseudorange prediction [23]. Then, the CP algorithms are extended by the 3DMA GNSS to achieve better performance in the urban area. The predicted satellite visibilities of neighboring road agents from the 3D building model are collaborated to achieve better position estimation [24,25]. Moreover, a complementary integration of the 3DMA GNSS ray-tracing technique and the DD based GNSS CP is developed to eliminate the systematic error and mitigate the uncorrelated NLOS error simultaneously [26]. The integration of 3DMA GNSS and CP has excellent potential to achieve satisfactory positioning accuracy in the urban area.

However, most of the CP algorithms are validated through naive simulated measurements, which behave very differently from the real experiments. A popular approach is to test the algorithm with additional simulated errors based on statistical models [20,27,28]. Although the normal distribution can model many of the measurement noises, the GNSS measurement error in the urban environment is dominated by the enormous signal delay due to building reflection. This delay is uniquely related to the geometry between satellites, GNSS receiver, and the reflecting surface, which may differ between each road agent. Hence, it is inappropriate to validate the CP algorithm with the GNSS measurement error simulated by the normal distribution or other statistical distribution models. Some studies directly use real measurements collected from multiple receivers for validation [19,26], whereas the agent amount is limited to a few. It is tough to test large-scale performance with large numbers of agents or study the influence of network size. Therefore, it is necessary to develop a realistic GNSS measurement simulator to study the CP performance in the urban area comprehensively.

There exist various types of GNSS measurement simulators, from tracking-level to measurement-level. The MUltipath Simulator Taking into Account Reflection and Diffraction (MUSTARD) developed by the Jet Propulsion Laboratory (JPL) is a famous GNSS measurement simulator [29], considering the signal reflection and diffraction effect based on the ray-tracing algorithm. The Satellite Navigation Radio Channel Signal Simulator (SNACKS) from the German Aerospace Center (DLR) further considers the multipath channel effect during simulation [30]. Moreover, the commercial GNSS simulator SimGEN® + SE-NAV developed by the Spirent Communication not only uses ray-tracing to track all valid

reflection or diffraction signals, but also considers the interaction between those traced signals [31]. To reduce the simulation complexity, another approach is developed, which simulates the tracking-level measurements in urban based on reference data collected from software defined receiver (SDR) [32]. The above simulators employ complicated algorithms to simulate the GNSS tracking-level measurements, whereas most CP studies only employ GNSS measurement-level data. For simplification, other GNSS simulators focusing on measurement-level data have been developed, such as the GNSS carrier-to-noise ratio ($C/N_0$) simulation for positioning aiding [33], the GNSS LOS/NLOS pseudorange simulation by ray-tracing [23], or the Doppler shift modeling under multipath effect [34,35]. However, many of them are not sophisticated enough with comprehensive error and noise modeling, for example, neglecting the multipath interaction error on pseudorange, or the tracking loop noise related to $C/N_0$. Moreover, those GNSS simulators are designed for one specific kind of measurement of a single receiver. It is also necessary to develop a simulator supporting multiagent full GNSS raw measurement simulation for CP studies.

In this study, a realistic GNSS measurement simulator is developed for multiple agents in the urban environment. Based on the satellite ephemeris, 3D building model, and the location of different road agents, the GNSS measurement-level data corresponding to each agent is simulated through the ray-tracing algorithm, including pseudorange, $C/N_0$, and Doppler frequency. To make the simulator suitable for CP studies, the locations of road agents are generated based on the dynamics and transportation mobility of road agents by SUMO (Simulation of Urban MObility) [36]. By considering the direct, reflected, diffracted, and multipath GNSS signal through ray-tracing, the measurement-level GNSS data with sophisticated modeling noises are simulated for multiple road agents in the urban area, in order to supply realistic large scale data for CP research. The contributions of this study are summarized as follows:

(1) Providing multiagent GNSS simulation for ITS and CP applications considering the transportation mobility from SUMO. By applying with different advanced positioning methods, the simulated measurements are capable of reflecting the positioning difficulties in the dense urban area.

(2) Holistically organizing and combing the important findings related to GNSS simulation from the related works and providing a complete GNSS simulator structure with detailed steps. A realistic measurement-level GNSS simulator is developed based on comprehensive error models without referencing RF data from SDR.

(3) A sophisticated GNSS simulator is developed for the GPS/BDS measurements in an urban scenario, in which the effects of signal reflection, diffraction and the interferences in-between are considered. The detailed modeling of those interferences on the pseudorange, $C/N_0$, and Doppler frequency measurements is presented and verified by comparing with real measurements.

The rest of the paper is structured as follows: the overall structure of the proposed realistic multiagent urban GNSS simulator is demonstrated in Section 2. The detailed procedures of ray-tracing simulation on direct, reflected, and diffracted GNSS signals are explained in Section 3. In Section 4, the GNSS measurement simulation will be elaborated based on different cases, including direct propagation, reflection, diffraction, and multipath. The modeling of GNSS measurement systematic error and noise are also explained in this section. Then, the performance of the proposed simulation is validated through different real experimental data and the positioning result based on popular positioning algorithms in Section 5. After discussing results and the future works in Section 6, the conclusion is drawn in Section 7.

## 2. Simulator Structure

The overall structure of the proposed GNSS realistic urban multiagent simulator (RUMS) is shown in Figure 1. Table 1 shows the key models and their corresponding references being employed in the simulator developed in this paper. First, the locations or trajectories of all the involving road agents require to be generated before GNSS measure-

ment simulation. For large scale CP studies, the transportation environment could influence the traffic flow and the available collaborator in a specific range. Therefore, it would be more realistic to evaluate the CP performance based on the simulated measurement considering the traffic condition. SUMO is an urban transportation simulator generating the trajectories of multiple road agents based on the transportation environment and facilities in a certain area. The traffic flow, road agent type (pedestrian or automobile), and other advanced settings can also be customized. In this study, we employ the simulated position $\mathbf{x}_R$ and velocity $\mathbf{v}_R$ of each road agent on GPS time from SUMO, namely the Road Agent PVT Data, to consider the urban mobility effect during the GNSS measurement simulation. Besides the information of road agents, the satellite positions $\mathbf{x}_{SV}$ from ephemeris and the 3D building model with building corner positions $\mathbf{x}_B$ are also required before simulation. The last required information is the Receiver Parameter/Model representing the hardware characteristic of the GNSS receiver in the simulation, which will be interpreted with details in Section 4.6.

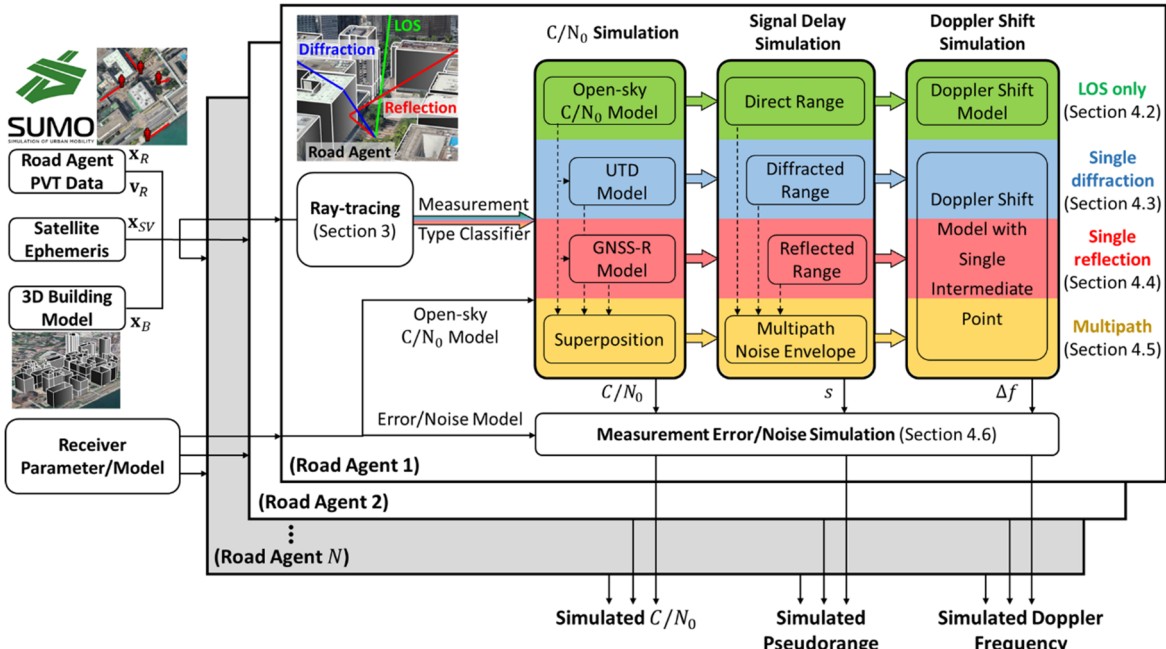

**Figure 1.** The flowchart of the proposed global navigation satellite system (GNSS) realistic urban multiagent simulator, GNSS RUMS.

**Table 1.** The employed models and the corresponding references in the proposed simulator.

| Model | Subsection in This Paper | Reference |
|---|---|---|
| Ray-tracing for the diffracted signal | Section 3.2 | [37] |
| Ray-tracing for the reflected signal | Section 3.3 | [23] |
| Open-sky $C/N_0$ regression model | Section 4.2 | [33] |
| UTD model | Section 4.3 | [38–40] |
| GNSS-R model | Section 4.4 | [41,42] |
| Multipath noise envelope | Section 4.5 | [43] |
| Doppler shift model with intermediate point | Sections 4.3 and 4.4 | [34] |
| Tracking loop noise model (DLL and FLL) | Section 4.6.1 | [44] |
| $C/N_0$ estimation noise (narrow-to-wide power method) | Section 4.6.2 | [45] |
| Satellite clock bias model | Section 4.6.3 | [44] |
| Receiver clock bias model (TCXO) | Section 4.6.4 | [46–48] |

**Table 1.** *Cont.*

| Model | Subsection in This Paper | Reference |
|---|---|---|
| Ionospheric delay (Klobuchar model) | Section 4.6.5 | [49] |
| Tropospheric delay (Saastamoinen model) | Section 4.6.6 | [50] |
| Elevation-based tropospheric delay variance model | Section 4.6.7 | [51] |
| Pseudorange user-equivalent-range-error budget | Section 4.6.7 | [44] |

For each road agent, the ray-tracing simulation is firstly employed based on the agent and satellite position as well as the 3D building model, in order to search all valid GNSS signal rays via direct propagation, diffraction or reflection from each satellite. Based on the ray-tracing result, the measurement of each satellite will be categorized into four types following different simulation strategies:

(1) LOS only: only the line-of-sight (LOS) signal propagation path is available and unobstructed;
(2) Single diffraction: only one diffraction path is available;
(3) Single reflection: only one reflection path is available; and
(4) Multipath: more than one propagation path is available, including any combination of the three cases above.

For the case of LOS only, the $C/N_0$ is simulated from an elevation-based open-sky $C/N_0$ regression model. The ranging measurement is the direct distance between satellite and agent without delay. The Doppler shift measurement is simulated from the Doppler effect model based on the location and velocity of the satellite and receiver.

For the single diffraction case, the $C/N_0$ measurement is simulated by the open-sky model and the uniform geometrical theory of diffraction (UTD) model [38]. The corresponding ranging measurement is the summation of the direct range and the extra delay due to diffraction. The diffracted Doppler shift is modeled by the Doppler effect with an intermediate point [34].

For the single reflection case, the $C/N_0$ is simulated based on the open-sky model with an attenuation factor from the GNSS reflectometer (GNSS-R) model [42,52]. The ranging measurement is modeled by adding the reflection delay on the direct distance. Similar to diffraction, the Doppler shift measurement of a reflected signal is simulated by the Doppler effect with an intermediate point.

For the case of multipath, the $C/N_0$ is simulated based on the superposition of fields from each valid signal, considering the interaction due to the differences in phase and amplitude. The corresponding ranging measurement is simulated by the multipath noise envelope based on the strength and delay of each signal, which takes the interferences in-between into account. The multipath Doppler shift is simulated based on the dominating signal path with the highest $C/N_0$ among all available paths.

Finally, the measurement systematic error and noise are simulated on top of the preceding $C/N_0$, ranging and Doppler shift simulation result based on the GNSS receiver characteristics. The $C/N_0$ measurement noise is simulated based on the standard deviation of the narrow-to-wide power ratio evaluation method [45]. The pseudorange measurement is simulated by incorporating the receiver clock bias, tropospheric delay, ionospheric delay, ephemeris error, satellite clock error, and a normally distributed noise based on the pseudorange error budget [44] for the above terms. Moreover, the $C/N_0$-related delay lock loop (DLL) noise is considered during the pseudorange simulation. The frequency lock loop (FLL) noise is also added on the preceding simulated Doppler shift. The simulated GNSS measurements of different agents are collected along GPS time and used for the CP algorithm test.

## 3. GNSS Ray-Tracing Simulation

In an urban scenario, the GNSS measurement is mainly degraded due to three effects, the signal blockage by buildings, the diffraction effect when the direct signal path is close to the building edge, and the reception of reflected signals from buildings. Therefore, three types of satellite signals can be received: the LOS signal, the diffracted signal, and the reflected signal, as shown in Figure 2, respectively. Based on the ray-tracing technique [53], all three types of signal can be traced with the geometrical relationship between buildings, satellite, and receiver.

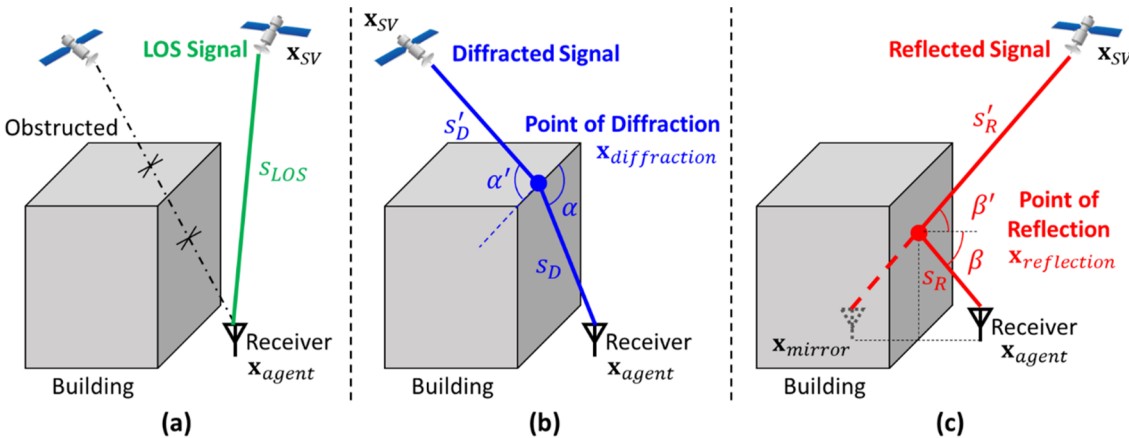

**Figure 2.** The valid GNSS signal tracked by ray-tracing algorithm, including: (**a**) the LOS signal; (**b**) the diffracted signal; (**c**) the reflected signal.

### 3.1. LOS GNSS Signal

Assuming the GNSS signal propagation follows a straight line, the GNSS LOS signal of a specific satellite can be directly traced as a line from the satellite position to the receiver position, as Figure 2a shows. Since the buildings in an urban area may obstruct the GNSS signal, only the LOS signal without any intersection with the building surfaces from the 3D building model will be considered as the valid LOS signal.

### 3.2. Diffracted GNSS Signal

The GNSS signal is an electromagnetic wave broadcasting from the satellite to ground. The propagation of waves towards a specific receiver may be bent when its straight transmitting path is slightly obstructed by the building, namely the diffraction effect. More specifically, the GNSS diffraction can also be regarded as the reception of the secondary wave emitted from the building edge. A detailed explanation of the GNSS diffraction phenomenon can be found in [40].

Based on the UTD [39], the GNSS diffraction can be modeled as an effect dominated by a specific diffracted signal, as Figure 2b, where the signal propagates along a bent path via the point of diffraction on the building edge. A valid diffraction path follows the Fermat's principle that the angle $\alpha'$ between incidence and the building edge is equal to the angle $\alpha$ between the diffracted ray and the edge [37]. By searching the point $\mathbf{x}_{diffraction}$ on the edges of the 3D building model that fulfill such geometrical relationship between satellite and receiver, the ray-tracing technique can be extended to trace all valid paths of the GNSS diffracted signal. In this study, we only consider the single diffraction effect and the diffracted signal being obstructed by other buildings is invalid.

### 3.3. Reflected GNSS Signal

Besides direct propagation and diffraction, the GNSS signal can also be reflected by the building surface and then received by the road agent, as Figure 2c shows. The reflected signal follows the Fermat's principle that the incident angle $\beta'$ is equal to the reflected angle

$\beta$. Based on this geometrical relationship, the ray-tracing technique finds the reflected signal path by searching for valid point of reflection $\mathbf{x}_{reflection}$ on all the surfaces in the 3D building model.

In detail, the receiver (road agent) location is firstly mirrored with respect to a specific building surface and denoted as $\mathbf{x}_{mirror}$. The point of reflection satisfying the geometrical relationship is located at the intersection between that surface and the line from the satellite to the mirrored location. If the point of reflection is within that mirroring surface and there is no obstruction from other building surfaces along the path, the corresponding path from the satellite passing the point of reflection to the receiver is a valid reflected signal. By applying this searching process to all the surfaces determined by the corner locations in the 3D building model, all valid reflected GNSS signal paths can be traced.

## 4. GNSS Measurement Simulation

After searching all the valid direct, diffracted, and reflected GNSS signals from the ray-tracing, the measurement status of each satellite is categorized into four types: LOS only, single reflection, single diffraction, and multipath. Different strategies are employed during the GNSS $C/N_0$, pseudorange, and Doppler frequency measurement simulation based on the measurement status. Finally, additional measurement error and noise based on comprehensive models are added on top of the preceding simulation result to make the final simulated measurement realistic.

### 4.1. Overall GNSS Measurement Simulation

To simulate the GNSS measurement realistically, all the error components need to be considered with appropriate models. For the $C/N_0$ measurement, besides the simulated value based on the signal strength, its hardware-related estimation noise also needs to be considered. The final simulated GNSS $C/N_0$ measurement is obtained by

$$C/N_{0,sim} = 10 \log_{10} \left( 10^{\frac{C/N_{0,type}}{10}} + \eta_{c/n_0} \right) \tag{1}$$

where $C/N_{0,type}$ is the simulated $C/N_0$ corresponding to the signal type and $type \in \{LOS, diffraction, reflection, multipath\}$. $\eta_{c/n_0} \sim \mathcal{N}\left(0, \sigma^2_{c/n_0}\right)$ is a random variable following Gaussian distribution (denoted by $\mathcal{N}$) with zero bias and the $C/N_0$ estimation variance $\sigma^2_{c/n_0}$.

The final simulated GNSS pseudorange measurement is given by

$$\rho_{sim} = s_{type} - cdt_S + cdt_r + \epsilon_{iono} + \epsilon_{tropo} + \eta_{DLL} + \eta_{models} \tag{2}$$

where $s_{type}$ is the simulated GNSS ranging measurement corresponding to different signal type, $c$ is the speed of light, $dt_S$ is the satellite clock bias, $dt_r$ is the receiver clock bias. $\epsilon_{iono}$ and $\epsilon_{tropo}$ are the ionospheric delay and the tropospheric delay, respectively. $\eta_{DLL} \sim \mathcal{N}\left(0, \sigma^2_{DLL}\right)$ and $\eta_{models} \sim \mathcal{N}\left(0, \sigma^2_{models}\right)$ are random variables with the variance from the DLL tracking loop noise and the systematic error model noise, respectively. The final simulated Doppler shift measurement is obtained by

$$\Delta f_{sim} = \Delta f_{type} + \eta_{FLL} \tag{3}$$

where $\Delta f_{type}$ is the simulated GNSS Doppler shift corresponding to different signal types, and $\eta_{FLL}$ is a random variable that $\eta_{FLL} \sim \mathcal{N}\left(0, \sigma^2_{FLL}\right)$ with the variance from the FLL tracking loop noise. Then, all three kinds of popular-used GNSS measurements are simulated and ready to be applied with different algorithms for evaluation.

The following Section 4.2, Section 4.3, Section 4.4, Section 4.5 demonstrate the procedures of obtaining the signal-type-related measurement error terms, corresponding to the cases of LOS-only, single diffraction, single reflection, and multipath. Section 4.6 introduces other systematic error terms related to the atmosphere, the hardware characteristics, etc.

### 4.2. LOS Only Case

The GNSS $C/N_0$ measurement relates to the strength of the signal, where a higher $C/N_0$ usually represents the signal being healthy with less interference. In an open-sky scenario without obstructions between a satellite and a receiver, the $C/N_0$ corresponding to a satellite signal is closely related to its elevation angle [54]. Therefore, the $C/N_0$ of an unobstructed signal can be simulated based on a GNSS elevation-$C/N_0$ regression model from long period open-sky data [33]. In this study, the $C/N_0$ from GPS satellite, Beidou satellite on geosynchronous equatorial orbit (GEO), on inclined geosynchronous orbit (IGSO), and on medium Earth orbit (MEO) are simulated with different open-sky models as Figure 3 shows, in order to consider the behavior varies between different systems and operating orbits. First-order polynomial fitting is employed between the elevation angle and the $C/N_0$ with the unit of Hz before applying logarithm.

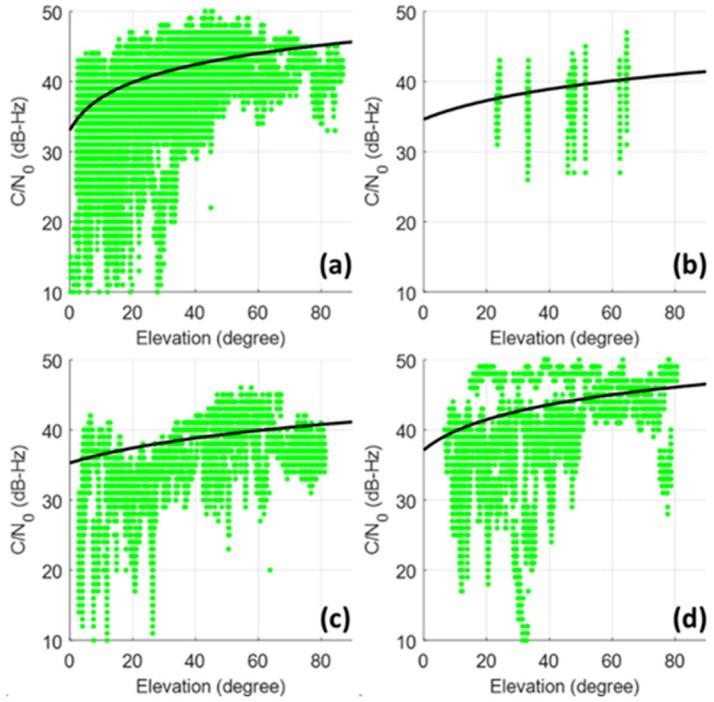

**Figure 3.** The open-sky elevation-$C/N_0$ regression model for satellites from different systems or orbits, including: (**a**) GPS satellites; (**b**) Beidou GEO satellites; (**c**) Beidou IGSO satellites; (**d**) Beidou MEO satellites. The green marker in each subfigure denotes the collected data from different satellites used for curve fitting. The black curve denotes the 1st order polynomial fitting model obtained from the $C/N_0$ data with the unit of Hz and visualized with the unit of dB-Hz. The green dots denote the $C/N_0$ data collected from a commercial grade receiver (Ublox EVK M8T) with a patch antenna in the open-sky environment.

The GNSS pseudorange measurement of the LOS only case is the direct range between satellite and road agent as follows

$$s_{LOS} = \| \mathbf{x}_{agent} - \mathbf{x}_{SV} \| \tag{4}$$

The Doppler frequency measurement of the LOS only case can be simulated based on the Doppler effect between two dynamic objects [34], using

$$\Delta f_{LOS} = \left( \mathbf{v}_{agent} \cdot \mathbf{a} - \mathbf{v}_{SV} \cdot \mathbf{a} - c \cdot \dot{t}_{agent} \right) / \lambda \tag{5}$$

where $\mathbf{v}_{agent}$ and $\mathbf{v}_{SV}$ are the velocity of the road agent and the satellite, respectively. $\mathbf{a}$ is the line-of-sight unit vector from the road agent to the satellite. $c$ is the speed of light and $\lambda$

is the wavelength of the corresponding satellite signal. $\dot{t}_{agent}$ is the receiver clock drift as a tuning parameter of the simulation.

### 4.3. Single Diffraction Case

For the satellite with only one diffracted signal path valid from the ray-tracing, the corresponding $C/N_0$, pseudorange, and Doppler measurements are simulated based on its geometrical behavior. Firstly, the $C/N_0$ is simulated based on the UTD, which describes the signal attenuation by the geometrical parameters of the diffracted signal and the building surfaces causing diffraction. The diffraction coefficient with respect to the strength of the LOS signal can be obtained from

$$\mathscr{D}_{UTD} = \frac{\mathscr{D}_{RR}}{\sqrt{s_D}} e^{-jks_D} \tag{6}$$

$$\mathscr{D}_{RR} = \frac{-\mathbf{u}_{\parallel}^d \mathbf{u}_{\parallel}^i \mathscr{D}_{\parallel} - \mathbf{u}_{\perp}^d \mathbf{u}_{\perp}^i \mathscr{D}_{\perp}}{2} \tag{7}$$

where $\mathscr{D}_{RR}$ is the attenuation factor between the right-hand circular polarization (RHCP) electric field before and after diffraction on the point of diffraction. $\mathscr{D}_{\parallel}$ and $\mathscr{D}_{\perp}$ are the soft and hard diffraction coefficients of a linear polarized field derived from the geometrical parameter of the diffracted signal and the building. $\mathbf{u}_{\parallel}^i$ and $\mathbf{u}_{\perp}^i$ denote the orthogonal components of the GNSS signal electric field parallel and perpendicular to the incident plane, respectively. Equation (7) extends the UTD from the linear polarized field to the RHCP field appropriate for the GNSS signal. $s_D$ is the distance between the point of diffraction and the receiver, $k$ is the GNSS signal wavenumber, and $j$ is the imaginary unit. $1/\sqrt{s_D}$ and $e^{-jks_D}$ denote the spreading factor and the phase shift of the signal emitted from the point of diffraction, respectively. The detailed procedures of computing the diffraction coefficient from the geometrical parameters can be found in [40]. Then, the $C/N_0$ of the diffracted signal can be obtained using

$$C/N_{0,diffraction} = C/N_{0,LOS} + 20\log|\mathscr{D}_{UTD}| \tag{8}$$

which is derived based on the relationships between the signal-to-noise ratio (SNR) with the diffraction coefficient and $C/N_0$ [55] as follows

$$\frac{SNR_{diffraction}}{SNR_{LOS}} = |\mathscr{D}_{UTD}|^2 \tag{9}$$

$$C/N_0 = 10\log SNR + f_{\text{BW}} \tag{10}$$

where $SNR_{LOS}$ and $SNR_{diffraction}$ denote the SNR peak power of the LOS and the diffracted signals, respectively. Equation (10) describes the relationship between $C/N_0$ (in the unit of dB-Hz) and SNR. $f_{\text{BW}}$ is the receiver front-end bandwidth which will be eliminated when considering only the power ratio between the LOS and diffracted signal. However, the $C/N_0$ of the original unobstructed LOS signal is difficult to obtain based on the receiver parameter that varies from different antennas. A convenient approach is to estimate the $C/N_{0,LOS}$ from an open-sky regression model, as the preceding LOS only case.

After obtaining the $C/N_0$ of the diffracted signal, the corresponding GNSS ranging measurement can be simulated by its total traveling distance using

$$s_{diffraction} = s_D' + s_D = \| \mathbf{x}_{diffraction} - \mathbf{x}_{SV} \| + \| \mathbf{x}_{agent} - \mathbf{x}_{diffraction} \| \tag{11}$$

where $s_D'$ is the distance between the satellite and the point of diffraction. The pseudorange error due to diffraction can be denoted as $\epsilon_D = s_{diffraction} - s_{LOS}$. Finally, the Doppler frequency of the diffracted signal can be derived based on the Doppler shift model extended with an intermediate point [34], using

$$\Delta f_{diffraction} = \left[ \left( 1 - \frac{\mathbf{v}_{SV} \cdot \mathbf{a}_D'}{c} \right) \mathbf{v}_{agent} \cdot \mathbf{a}_D - \mathbf{v}_{SV} \cdot \mathbf{a}_D' - c \cdot \dot{t}_{agent} \right] / \lambda \approx \left( \mathbf{v}_{agent} \cdot \mathbf{a}_D - \mathbf{v}_{SV} \cdot \mathbf{a}_D' - c \cdot \dot{t}_{agent} \right) / \lambda \tag{12}$$

where $\mathbf{a}'_D$ is the line-of-sight unit vector from the point of diffraction $\mathbf{x}_{diffraction}$ to the satellite and $\mathbf{a}_D$ is the unit vector from the road agent to the point of diffraction. Based on the position and velocity of the road agent and the satellite as well as the location of the point of diffraction, the Doppler frequency of the GNSS signal with a specific wavelength can be simulated correspondingly.

### 4.4. Single Reflection Case

The GNSS-R has been widely employed for geodetic survey application, which estimates the geometrical parameter during reflection based on the reflected $C/N_0$ measurement. The mechanism of GNSS-R can be, in turn, used to simulate the GNSS $C/N_0$ measurement based on the geometrical parameters during reflection. The $C/N_0$ relationship between the LOS signal and the reflected signal can be derived by

$$C/N_{0,reflection} = C/N_{0,LOS} + 20log\left(\frac{s_{LOS}}{s'_R + s_R}|\mathscr{R}_{LR}|\right) \tag{13}$$

based on Equation (10) and the relationship that

$$\frac{SNR_{reflection}}{SNR_{LOS}} = \frac{s_{LOS}^2}{\left(s'_R + s_R\right)^2}|\mathscr{R}_{LR}|^2 \tag{14}$$

where $s'_R$ is the distance between the satellite and the point of reflection and $s_R$ is the distance between the point of reflection and the receiver (road agent). Equation (13) is derived by Equations (10) and (14), which describe the reflected/LOS SNR peak power ratio [56] and the conversion from $C/N_0$ (in the unit of dB-Hz) to SNR, respectively. Therefore, the $C/N_0$ of a reflected signal can be represented by its LOS $C/N_0$ attenuated by a spreading factor and a reflection coefficient $\mathscr{R}_{LR}$. Note that the GNSS signal is RHCP, whereas the reflected GNSS signal may be inverted into left-hand circular polarization (LHCP). By considering the polarization interference, the reflection coefficient $\mathscr{R}_{LR}$ from RHCP to LHCP can be written as the combination of the reflection coefficient on horizontal and vertical polarization [41,57], as follows

$$\mathscr{R}_{LR} = \frac{1}{2}(\mathscr{R}_{vv} - \mathscr{R}_{hh}) \tag{15}$$

where $\mathscr{R}_{hh}$ and $\mathscr{R}_{vv}$ are the Fresnel reflection coefficient of the horizontal and vertical linear polarization component with respect to the reflecting surface, derived from

$$\mathscr{R}_{hh} = \frac{\cos\beta' - \sqrt{\varepsilon - \sin^2\beta'}}{\cos\beta' + \sqrt{\varepsilon - \sin^2\beta'}} \tag{16}$$

$$\mathscr{R}_{vv} = \frac{\varepsilon\cos\beta' - \sqrt{\varepsilon - \sin^2\beta'}}{\varepsilon\cos\beta' + \sqrt{\varepsilon - \sin^2\beta'}} \tag{17}$$

$$\varepsilon = \varepsilon_2/\varepsilon_1 \tag{18}$$

$\beta'$ is the incident angle of the reflected signal, $\varepsilon_1$ is the dielectric constant of the air, and $\varepsilon_2$ is the dielectric constant of the reflecting surface. In this study, the dielectric constant of glass ($\varepsilon_2 = 4.7$) is used for the reflector, and the conduction due to the lossy medium is neglected. Note that some articles denote $\mathscr{R}_{hh}$ by $\mathscr{R}_\perp$ and $\mathscr{R}_{vv}$ by $\mathscr{R}_\parallel$ as the linear polarized component perpendicular and parallel to the incident plane (the plane determined by the incident and reflected signals), which are also representing the soft and hard reflections, respectively. Based on the geometrical parameter, including the incident angle $\theta$ and the distances between satellite, road agent, and the point of reflection, the $C/N_0$ of the

corresponding reflected GNSS signal can be simulated from its unobstructed $C/N_{0,LOS}$ by the open-sky model.

The ranging measurement of single reflected GNSS signal can be simulated by the total travelling distance on the reflection path, as follows.

$$s_{reflection} = s'_R + s_R = \parallel \mathbf{x}_{reflection} - \mathbf{x}_{SV} \parallel + \parallel \mathbf{x}_{agent} - \mathbf{x}_{reflection} \parallel \tag{19}$$

The corresponding pseudorange delay due to reflection can be written as

$$\epsilon_R = s_{reflection} - s_{LOS} \tag{20}$$

The Doppler frequency of the reflected signal can be derived based on the Doppler shift model extended with an intermediate point [34], using

$$\begin{aligned}
\Delta f_{reflection} &= \left[ \left( 1 - \frac{\mathbf{v}_{SV} \cdot \mathbf{a}'_R}{c} \right) \mathbf{v}_{agent} \cdot \mathbf{a}_R - \mathbf{v}_{SV} \cdot \mathbf{a}'_R - c \cdot \dot{t}_{agent} \right] / \lambda \\
&\approx \left( \mathbf{v}_{agent} \cdot \mathbf{a}_R - \mathbf{v}_{SV} \cdot \mathbf{a}'_R - c \cdot \dot{t}_{agent} \right) / \lambda
\end{aligned} \tag{21}$$

where $\mathbf{a}'_R$ is the line-of-sight unit vector from the point of reflection $\mathbf{x}_{reflection}$ to the satellite, and $\mathbf{a}_R$ is the unit vector from the road agent to the point of reflection. Based on the position and velocity of the road agent and the satellite as well as the location of the point of reflection, the Doppler frequency of the GNSS signal with a specific wavelength can be simulated correspondingly.

*4.5. Multipath Case*

Besides the reception of a single GNSS signal from LOS, reflection, or diffraction, the GNSS receiver can also receive multiple signals simultaneously, if available, namely the multipath case. The corresponding measurement will be degraded due to the interferences between each signal. In this study, we only consider the interaction between the shortest two signals with the attenuation less than a heuristically designed threshold of 20 dB-Hz. Other available signals with larger delay or lower strength have fewer effects and they are neglected. Therefore, the multipath case contains the signal combination of: LOS and reflection, LOS and diffraction, reflection and diffraction, double reflections, and double diffractions.

Based on the expression of individual fields, the expression of the joint signal field with interferences between signals can be derived through the superposition of fields. The detailed derivation with different signal combination cases is demonstrated in Appendix A. Then, the strength ratio between the multipath joint field and the unobstructed field on the road agent location can be expressed by

$$\Gamma_{multipath} = \Gamma_a + \Gamma_b \tag{22}$$

$$\Gamma_{a,b} = \begin{cases} 1, & a, b \in LOS \\ \frac{\mathcal{D}_{RR}}{\sqrt{s_D}} e^{-jk\epsilon_D}, & a, b \in diffraction \\ \mathcal{R}_{LR} e^{-jk\epsilon_R}, & a, b \in reflection \end{cases} \tag{23}$$

where $\Gamma_a$ and $\Gamma_b$ represent the stand-alone signal strength ratio compared to the unobstructed signal for different cases, $\epsilon_D$ and $\epsilon_R$ denote the extra traveling distance compared to the unobstructed signal for the diffracted signal or the reflected signal, respectively. Analogous to the single reflection case or single diffraction case with the approximation the extra delay due to building interference is negligible compared to the total range from satellite to receiver, the $C/N_0$ simulation of the multipath case can be obtained from

$$C/N_{0,multipath} = C/N_{0,LOS} + 20 log \left| \Gamma_{multipath} \right| \tag{24}$$

On the other hand, the GNSS pseudorange measurement under the multipath case is also affected by the interaction between signals. The code pseudorange error due to this interaction needs to be evaluated by the multipath noise envelope model with the amplitude ratio and the delay between two signals. Based on the multipath noise envelope for the early-minus-late power discriminator [43], the multipath error on the pseudorange can be simulated by

$$\epsilon_{multipath} = F_{multipath \ noise \ envelope}(\alpha, \beta, \Delta\tau, d) \tag{25}$$

$$\alpha = 10^{\frac{C/N_{0,b} - C/N_{0,a}}{20}} \tag{26}$$

$$\beta = -\frac{2\pi}{\lambda}\Delta s \tag{27}$$

$$\Delta\tau = \Delta s / L_{chip} \tag{28}$$

where $F_{multipath \ noise \ envelope}$ denotes the multipath error estimation function based on the multipath noise envelope demonstrated in Figure 4. $\alpha$ denotes the signal amplitude ratio ($\alpha \leq 1$ for this case), $\beta$ denotes the carrier phase offset and between two signals. $d$ is the time spacing between early and late correlator. $\Delta s$ and $\Delta\tau$ are the multipath time and distance delay in-between, respectively. $L_{chip}$ is the code chip width. In this study, the carrier phase shift induced by the reflector is neglected.

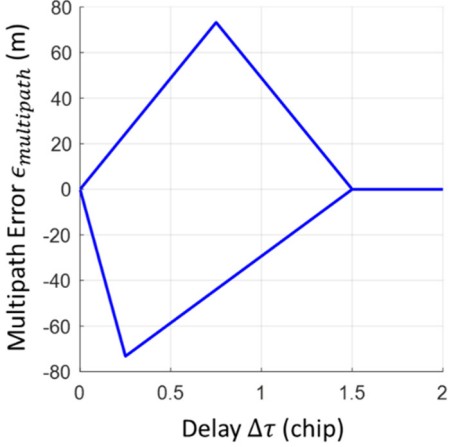

**Figure 4.** The demonstration of the multipath noise envelope when $d = 1$ (chip) and the delayed signal has half amplitude attenuation compared to the early signal.

Then, the GNSS code pseudorange measurement can be obtained by

$$s_{multipath} = s_{LOS} + \epsilon_{multipath} \tag{29}$$

For the Doppler effect, it is assumed to be dominated by the strongest signal [35], which has the highest $C/N_0$. Hence, the Doppler shift for the multipath case can be simulated by the dominated signal through Equations (5), (12), or (21), as follows.

$$\Delta f_{multipath} = \begin{cases} \Delta f_{LOS}, & LOS \ signal \ dominated \\ \Delta f_{diffraction}, & diffracted \ signal \ dominated \\ \Delta f_{reflection}, & reflected \ signal \ dominated \end{cases} \tag{30}$$

### 4.6. General Measurement Error and Noise

To ensure the GNSS measurement simulation is realistic, other measurement errors and noises need to be considered besides the preceding building-related errors. In this study, sophisticated models are employed to simulate different kinds of error and noise

generally appearing in GNSS measurements, including the tracking loop noise, $C/N_0$ estimation noise, satellite clock bias, receiver clock error, ionospheric delay, tropospheric delay, and pseudorange error modeling noise.

4.6.1. Tracking Loop Noise

The pseudorange measurement noise from DLL consists of the thermal noise and the dynamic stress [44]. The thermal noise is related to the signal $C/N_0$ and the receiver parameters, whereas the dynamic stress is related to the loop order and bandwidth. This paper follows [44] to generate the tracking loop noise. Since the dynamic stress can be almost removed via the carrier aided code technique, it is neglected during the DLL noise simulation. As a result, the 1-sigma DLL noise of C/A code under BPSK-R modulation is obtained by

$$\sigma_{DLL} \approx \sigma_{tDLL} \cong \begin{cases} \sqrt{\dfrac{B_{code}}{2C/N_0} d \left[1 + \dfrac{2}{T_i C/N_0(2-d)}\right]}, d \geq \dfrac{\pi R_c}{B_{fe}} \\[3ex] \sqrt{\dfrac{B_{code}}{2C/N_0} \left[\dfrac{1}{B_{fe}T_c} + \dfrac{B_{fe}T_c}{\pi - 1}\left(d_{e-l} - \dfrac{1}{B_{fe}T_c}\right)^2\right] \times \left[1 + \dfrac{2}{T_i C/N_0(2-d)}\right]}, \dfrac{R_c}{B_{fe}} < d < \dfrac{\pi R_c}{B_{fe}} \\[3ex] \sqrt{\dfrac{B_{code}}{2C/N_0} \left(\dfrac{1}{B_{fe}T_c}\right)\left[1 + \dfrac{1}{T_i C/N_0}\right]}, d \leq \dfrac{R_c}{B_{fe}} \end{cases} \quad (31)$$

where $\sigma_{tDLL}$ is the 1-sigma thermal noise code tracing jitter, $d$ is the early-to-late correlator spacing (unit of chip), $B_{fe}$ is the front-end bandwidth, $R_c$ is the chip rate, $B_{code}$ is the code loop noise bandwidth, $T_i$ is the predetection integration time, and $T_c$ is the chip period. The 1-sigma Doppler measurement noise $\sigma_{FLL}$ from a 2nd order FLL consists of the thermal noise frequency jitter $\sigma_{tFLL}$ and the dynamic stress term $f_e$ [44], which can be estimated as follows (in the unit of Hz).

$$\sigma_{FLL} = \sigma_{tFLL} + \frac{f_e}{3} \quad (32)$$

$$\sigma_{tPLL} = \frac{1}{2\pi T_i} \sqrt{\frac{4F B_{carrier}}{C/N_0}\left(1 + \frac{1}{T_i C/N_0}\right)} \quad (33)$$

$$f_e = \frac{d^3 R/dt^3}{360\omega_0^3} \quad (34)$$

where $F$ equals 1 for high $C/N_0$ and equals 2 for the $C/N_0$ close to the threshold $1/(12T_i)$. $\omega_0 = B_{carrier}/0.53$ is the loop filter natural radian frequency.

Figure 5 shows an example of the 1-sigma noise distribution with respect to the $C/N_0$ value on measurements corresponding to DLL and FLL tracking loop. Normally, a high $C/N_0$ indicates the signal is healthy, whereas a low $C/N_0$ indicates the signal has larger noise during the tracking loop. Therefore, the tracking loop noises on the GNSS pseudorange and Doppler shift measurement are modeled by the random variables following zero-bias Gaussian distribution with the corresponding standard deviation.

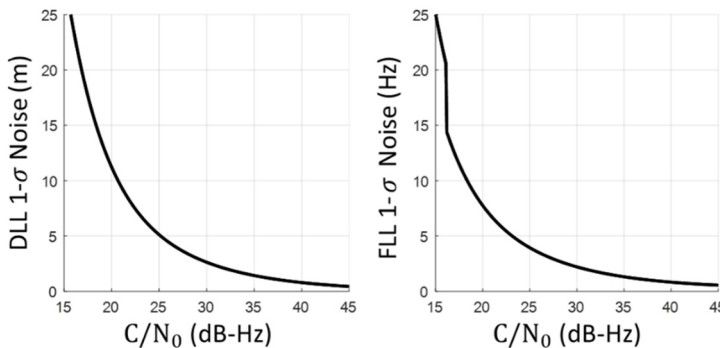

**Figure 5.** The 1-sigma measurement noise on DLL and FLL corresponding to the $C/N_0$ value.

### 4.6.2. $C/N_0$ Estimation Noise

For real cases, the GNSS $C/N_0$ could also be noisy due to interferences or weak signal power. During the narrow-to-wide power method, a popular approach for $C/N_0$ estimation, the $C/N_0$ measurement noise can be evaluated statistically in the form of standard deviation [45].

$$\sigma_{c/n_0} \approx \frac{1}{T_{coh}} \frac{M-1}{\left(M - \overline{P}_{N/W}\right)^2} \frac{\sigma(P_N/P_W)}{\sqrt{K}} \tag{35}$$

where the variables $K$ and $M$ denote that the prompt I and Q are divided into $K$ intervals with $M$ samples in each interval for the $C/N_0$ estimation. $T_{coh}$ is the coherent integration time, $\overline{P}_{N/W}$ and $\sigma(P_N/P_W)$ are the averaged value and standard deviation of the narrow and wide power measurements ratio on $K$ intervals, respectively. However, $\overline{P}_{N/W}$ and $\sigma(P_N/P_W)$ are obtained from the tracking-level measurements not available in the proposed simulator. An alternative approach is to model these two parameters by curve fitting models with large amounts of data in urban scenarios. In this study, based on the $c/n_0$ value (which is the $C/N_0$ with the unit of Hz), a rational fitting model is employed for $\overline{P}_{N/W}$, whereas an exponential fitting model is employed for $\sigma(P_N/P_W)$, as Figure 6 shows. Then, $\overline{P}_{N/W}$ and $\sigma(P_N/P_W)$ can be modeled based on the simulated $C/N_0$ value, and further combined with the receiver parameter $T_{coh}$, $M$ and $K$ to simulate the noise on the $C/N_0$ measurement.

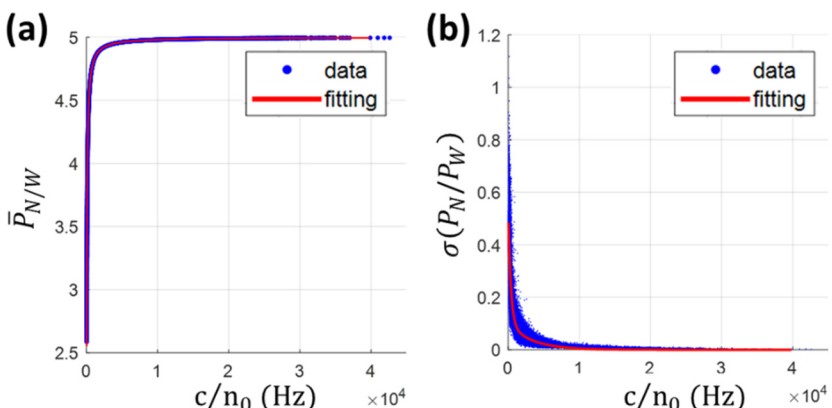

**Figure 6.** The curve fitting model for $C/N_0$ noise simulation based on large amount data in the urban scenario: (**a**) $\overline{P}_{N/W}$ fitted by a rational model; (**b**) $\sigma(P_N/P_W)$ fitted by an exponential model.

### 4.6.3. Satellite Clock Bias

The actual GNSS pseudorange measurement contains the satellite clock error due to imperfect synchronization with the corresponding system time. The correction for this error is estimated by the control segment and broadcast to users through the navigation message

from satellite. Therefore, this error term can be obtained for the pseudorange simulation based on the correction parameters from the satellite ephemeris [44,58], as follows.

$$dt_S = a_{f0} + a_{f1}\left(t - t_{eph}\right) + a_{f2}\left(t - t_{eph}\right)^2 + \Delta t_r \tag{36}$$

$$\Delta t_r = Fe\sqrt{A}sinE_k \tag{37}$$

where $t$ is the receiver current time, $t_{eph}$ is the ephemeris reference time, and $\Delta t_r$ is the relativistic correction term. $a_{f0}$, $a_{f1}$, $a_{f2}$, $e$, and $A$ are the clock bias, clock drift, frequency drift, satellite orbital eccentricity, and orbit semimajor axis obtained from the ephemeris data, respectively. $F = -4.442807633 \times 10^{-10}\left(s/m^{1/2}\right)$ is the constant and $E_k$ is the satellite orbit eccentric anomaly.

### 4.6.4. Receiver Clock Bias

Besides the satellite clock bias, the other term making the GNSS ranging measurement 'pseudo-range', is the receiver clock bias, which is usually an unknown variable needed to be estimated during positioning. In a real case, the receiver clock bias is not stable and drifts variously, related to the referencing oscillator of the receiver. A practical approach to simulate the receiver clock bias $dt_r$ with drift is through a 1st order linear approximation model based on the receiver parameters [46,47], as follows.

$$dt_r = dt_{r,t0} + \Delta dt_r(t - t_0) \tag{38}$$

$$\Delta dt_r = \frac{\Delta f_0 + \varepsilon_{RWFM} + \varepsilon_{FFM} + \varepsilon_{WFM}}{f_0} \tag{39}$$

where $dt_{r,t0}$ is the initial clock bias of the receiver on time $t_0$, $t$ is the receiver current time. $\Delta dt_r$ is the frequency offset determined by the initial frequency offset $\Delta f_0$, random walk frequency modulation noise $\varepsilon_{RWFM}$, flicker frequency modulation noise $\varepsilon_{FFM}$, white frequency modulation noise $\varepsilon_{WFM}$, and the local oscillator frequency $f_0$. Figure 7 shows an example of the clock errors from $\varepsilon_{RWFM}$, $\varepsilon_{FFM}$, and $\varepsilon_{WFM}$ based on the TCXO parameters from [48].

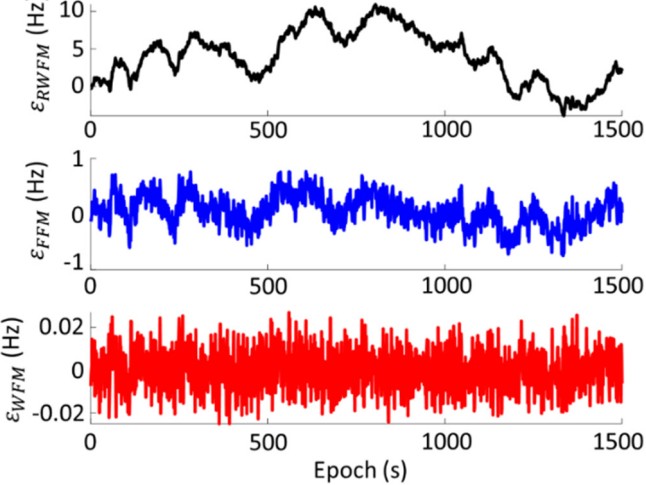

**Figure 7.** An example of the simulated values of random walk frequency modulation noise $\varepsilon_{RWFM}$, flicker frequency modulation noise $\varepsilon_{FFM}$, and white frequency modulation noise $\varepsilon_{WFM}$ based on the receiver parameters with TCXO from [48].

### 4.6.5. Ionospheric Delay

The real GNSS pseudorange measurement contains the error due to ionospheric delay, which is usually corrected by the Klobuchar model with detailed procedures in [49], and summarized as below.

$$\epsilon_{iono} = F\left[5 \times 10^{-9} + \left(\sum_{n=0}^{3} \alpha_n \Phi_m^n\right)\left(1 - \frac{x^2}{2} + \frac{x^4}{24}\right)\right] \tag{40}$$

$$x = \frac{2\pi(t - 50400)}{\sum_{n=0}^{3} \beta_n \Phi_m^n} \tag{41}$$

where $F$ is an elevation-related slant factor, $\alpha_n$ and $\beta_n$ are the ionospheric parameters from ephemeris data, $\Phi_m^n$ is the geomagnetic latitude, $t$ is the local time. Based on the receiver latitude and longitude, satellite elevation and azimuth angles, receiver time, and the ionospheric parameters from the ephemeris, the corresponding ionospheric delay can be simulated via the Klobuchar model correspondingly.

### 4.6.6. Tropospheric Delay

Another error term always contained in the pseudorange measurements is the delay due to tropospheric refraction, described by the Saastamoinen model [50], as follows.

$$\epsilon_{tropo} = 0.002277sec(el)[p_t + \left(\frac{1255}{T_{abs}} + 0.05\right)p_{wv} - Btan^2el] \tag{42}$$

where $el$ is the satellite elevation angle. $p_t$, $p_{wv}$, $T_{abs}$, and $B$ are the total barometer pressure, the partial pressure of water vapor, absolute temperature (in Kelvin), and the coefficient term related to the ellipsoid height of the receiver. Based on the satellite elevation angle and the receiver ellipsoid height, the corresponding tropospheric delay $\epsilon_{tropo}$ can be simulated correspondingly.

### 4.6.7. Pseudorange Error Modeling Noise

Besides the exact value of errors from different models, each error term may also have a certain variation during real operation. These variations of errors are caused by various unknown factors, which are difficult to model. A practical way to simulate these variations is according to the GNSS user-equivalent-range-error (UERE) budget from [44]. The nominal 1-sigma error of the satellite clock $\sigma_{SC}$ is given as 1.1 m. The 1-sigma ephemeris prediction error $\sigma_{eph}$ on pseudorange is given as 0.8 m. The noise on the relativistic effect error is negligible. The 1-sigma error on ionospheric delay $\sigma_{iono}$ is given by a higher bound as 7.0 m, which is reduced in half in this study by assuming a good ionospheric correction. The 1-sigma error on tropospheric delay is obtained from an elevation-angle-related model [51], as follows.

$$\sigma_{tropo} = 0.12\frac{1.001}{\sqrt{0.002001 + \sin^2 el}} \tag{43}$$

As a result, all the error noise above can be combined into a total error noise factor, as the error modeling noise $\sigma_{models}$ shown below.

$$\sigma_{models} = \sqrt{\sigma_{SC}^2 + \sigma_{eph}^2 + \sigma_{iono}^2 + \sigma_{tropo}^2} \tag{44}$$

Then, the total modeling noise on the pseudorange measurements is simulated by a random variable from zero-bias Gaussian distribution with the standard deviation of $\sigma_{models}$.

## 5. Simulation Performance Verification

In this section, the proposed realistic GNSS measurement simulator for multiple agents in the urban area is verified by two approaches. On the one hand, the simulated GNSS measurements of $C/N_0$, pseudorange and Doppler frequency are compared with the real experimental measurements collected from a commercial-grade receiver in different scenarios to verify whether different interferences can be appropriately simulated. On the other hand, the simulated GNSS measurements of different agents will be applied with conventional positioning algorithms and collaborative positioning algorithms, aiming to verify that simulated measurements can reproduce realistic performances of current positioning algorithms.

### 5.1. Experimental Verification

#### 5.1.1. Experiment Setup

The experimental verification of the proposed simulator consists of three scenarios: (1) one-hour static experiment on the intersection of the urban area; (2) dynamic vehicular experiment in the urban area; (3) static experiment with multiagent in different environments. For the verification using the one-hour static data, the experiment locations and the corresponding sky-plots with buildings are shown in Figure 8. The GNSS measurement on this location is expected to be severely degraded by reflection, diffraction, or multipath. For the dynamic experiment, the trajectory is shown in Figure 9, where the vehicle is moving from the open-sky area to the urban area, and then back to the starting point. The receiver locations and sky-plots during the multiagent static experiment are shown in Figure 10 with a summary of the surrounding environment of each receiver in Table 2.

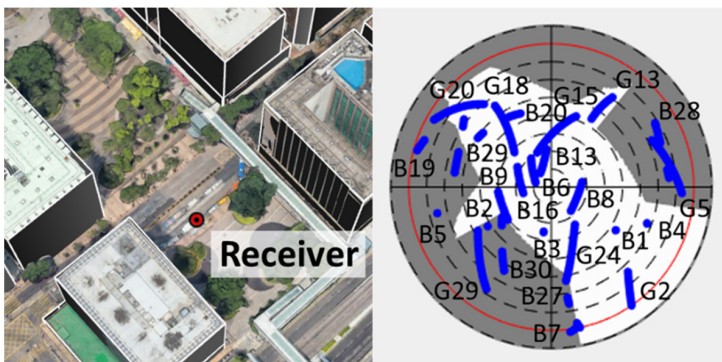

**Figure 8.** The location and the corresponding sky-plot with building boundaries for the one-hour static experiment on an intersection in an urban area. The right sky-plot demonstrates the satellite distribution and the building blockage (shaded area) on the experiment location (red marker).

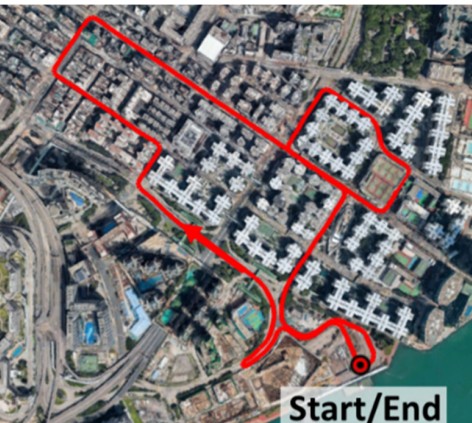

**Figure 9.** The trajectory during the dynamic vehicular experiment.

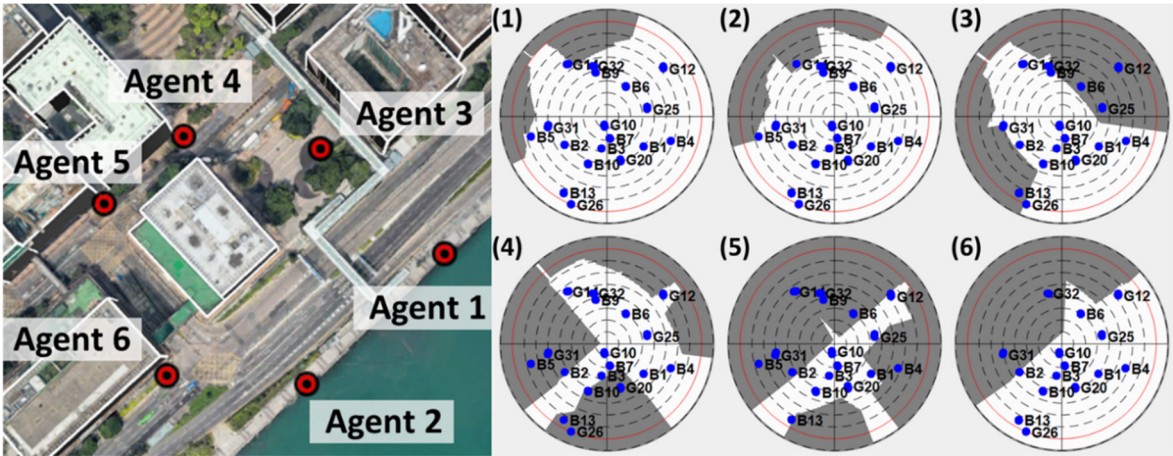

**Figure 10.** The receiver locations and the sky-plots during the multiagent static experiment. The sub-figures (**1**)–(**6**) are corresponding to the sky-plots with buildings and satellites for Agent 1–6.

**Table 2.** Surrounding environment of each receiver during the multiagent experiment.

| Receiver | Environment |
|---|---|
| 1 | Open-sky |
| 2 | Open-sky |
| 3 | Light urban area |
| 4 | Urban intersection |
| 5 | Dense urban area |
| 6 | Light urban area with one-side of building |

During all the experiments, the commercial-grade GNSS receiver ublox EVK-M8T with patch antenna is employed to collect GNSS (GPS and Beidou) raw measurements. The receiver true location of each static experiment is obtained based on the landmark on Google Earth, which has the accuracy within 1–2 m from our experience. The receiver true location during the dynamic experiment is obtained based on the GNSS/INS integrated solution from the Novatel SPAN-CPT capable of RTK positioning. The value of receiver parameters used in the proposed GNSS simulator are listed in Table 3. During the simulation, unlike the $C/N_0$ and Doppler frequency, the pseudorange measurement always contains the receiver clock bias, which is a receiver-related unknown. It is hard and unnecessary to simulate this bias exactly the same as the real measurement, since it is also estimated in the solution during positioning. Therefore, instead of comparing with the bias-contained pseudorange, we directly compare the pseudorange error between real measurement and simulation for verification. The pseudorange error in real measurement is labeled by applying the double different technique with the measurement from a nearby reference station [59]. Similarly, the real Doppler frequency measurement also contains a receiver clock drift that is difficult to exactly simulate. Therefore, we apply the single difference approach to estimate the Doppler shift error for the consistency evaluation between real measurement and simulation. The Doppler shift error of the $i^{th}$ satellite is estimated by

$$\varepsilon_f^i = \left( \Delta f_{agent}^i - \Delta f_{agent}^m \right) - \left( \Delta f_{LOS}^i - \Delta f_{LOS}^m \right) \tag{45}$$

where the superscript $m$ denotes the Doppler shift of the selected master satellite (with the highest elevation angle). The subscript *agent* denotes the Doppler shift from measurement or simulation. The subscript *LOS* denotes the true Doppler shift labeled using the user velocity, satellite velocity, and the satellite line-of-sight vector. By applying the single dif-

ference on the Doppler shift between the target satellite and the master satellite, the shared receiver clock drift can be eliminated, which provides a clearer consistency evaluation between the real received and the simulated Doppler shift.

**Table 3.** The value of receiver-related parameters during simulation.

| Notation | Parameter | Value | Notation | Parameter | Value |
|---|---|---|---|---|---|
| $B_{carrier}$ | Carrier loop noise bandwidth | 15 (Hz) | $T_{coh}$ | Coherent integration time | 0.02 (s) |
| $B_{code}$ | Code loop noise bandwidth | 0.2 (Hz) | $T_i$ | Predetection integration time | 0.02 (s) |
| $B_{fe}$ | Front-end bandwidth | 2.046 (MHz) | $\sigma_A$ | Allan deviation of the reference oscillator | $1 \times 10^{-10}$ |
| $d$ | Early-to-late correlator spacing | 1 (chip) | $\Gamma_{threshold}$ | Signal strength attenuation threshold | 20 (dB) |
| $K$ | Intervals of prompt I and Q during $C/N_0$ estimation | 10 | $d^3R/dt^3$ | Maximum LOS jerk dynamics | 98 (m/s$^3$) |
| $M$ | Samples in each interval $K$ | 5 | - | - | - |

### 5.1.2. One-Hour Static Experiment Performance Verification

During the static experiment on the urban intersection, the GNSS measurements from each satellite are simulated by the proposed simulator (please see Supplementary Materials) and compared with the real received measurement. Figure 11 shows the simulation result of $C/N_0$, pseudorange error, and Doppler shift error from the B4 satellite, which is classified as a LOS satellite during the whole simulation. The simulated $C/N_0$ measurement is consistent with the real measurement, verifying the proposed open-sky regression model can be used to simulate the $C/N_0$ from the LOS satellites. For the pseudorange measurement, the simulation result is also consistent with the real measurement, which is a typical LOS measurement without large errors. Although the real measurement has less fluctuation compared to the simulation due to some filtering technique in receivers, both simulated and real measurements have a similar magnitude of noise, representing the behavior of the LOS measurement. For the Doppler shift, the error from simulation has a slightly higher noise compared to that from the real measurement, but both of them are in a similar level.

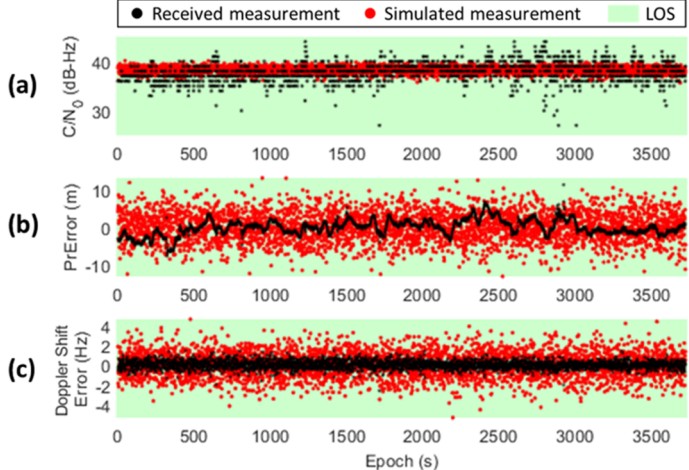

**Figure 11.** The comparison of the simulated and actual received GNSS measurements from B4 satellite during the one-hour static experiment, including: (**a**) $C/N_0$; (**b**) pseudorange error; (**c**) single-differenced Doppler shift error. The black marker and red marker denote the real and simulated measurement, respectively. The green background denotes the measurement type as LOS during the simulation.

The simulation results from the G20 satellite experiencing all four measurement status are compared with the real measurements in Figure 12. In the first 350 s, only one valid reflected signal path is found by the ray-tracing, which corresponds to the single reflection case in the simulator. The $C/N_0$ is simulated with a large attenuation to around 28 dB-Hz, whereas the pseudorange measurement is simulated with around 30 m delay. In this period, the real received measurement is consistent with the simulation, containing a similar $C/N_0$ attenuation and pseudorange delay. In the later period, the proposed simulator only finds one valid diffracted signal while the previous reflection becomes invalid, which turns the measurement status to single diffraction case. The $C/N_0$ is gradually increased, since the satellite elevation angle rises towards the building edge with a decreasing signal strength attenuation coefficient. On the other hand, the pseudorange errors in real measurement and simulation both are reduced since the diffraction delay is normally negligible comparing to other noise. After the 500 epoch, the satellite is raised from the building boundary and becomes visible to the receiver, making both diffracted and direct signal path valid to the receiver in the simulation. Therefore, the measurement is simulated with the multipath case. Consistent with the real measurement, the $C/N_0$ is continually increasing while the simulated pseudorange error fluctuates around zero. After the 1300 epoch, the diffracted signal path becomes invalid for the receiver, and only the direct signal remains. Similar to the real measurement, the $C/N_0$ is maintained in a high value and the simulated pseudorange error is dominated by zero-biased random noises. In the ending period, the real $C/N_0$ is attenuated compared to the simulation, but no large constant delay is found in the pseudorange error. This could possibly be due to the signal interferences from those miss-detected reflections.

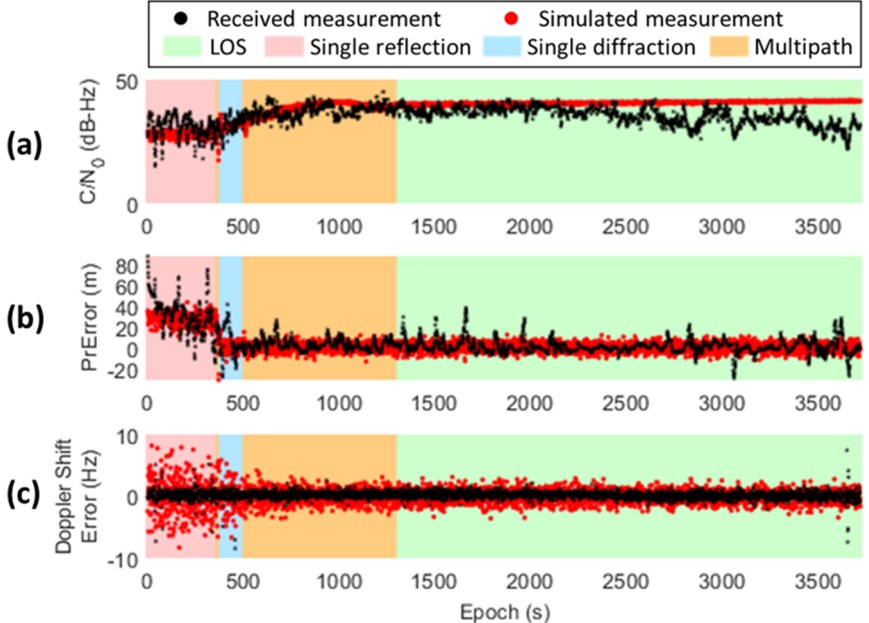

**Figure 12.** The comparison of the simulated and actual received GNSS measurements from G20 satellite during the one-hour static experiment, including: (**a**) $C/N_0$; (**b**) pseudorange error; (**c**) Doppler shift error. The colored background denotes the measurement status being categorized during the simulation, including: LOS, single reflection, single diffraction, and multipath.

For the satellite G29 with large pseudorange errors probably due to reflection, the corresponding simulated and real received measurements are shown in Figure 13. During the experiment, its real received $C/N_0$ measurement is fluctuated with a large amplitude. This is a typical consequence of the multipath effect, where the phase shift between two signal paths is rapidly switching between in-phase and out-of-phase due to the satellite movement. Moreover, the pseudorange error in the real received measurement also has

a large fluctuation but contains an enormous bias from zero. This is also another line of evidence of the multipath effect occurring, or more specifically, the multipath effect with a dominating reflected signal and no valid direct signal. The proposed simulator traces multiple valid propagation paths and correctly simulates the occurrence of multipath effect in the beginning (except the first 50 epochs) and ending period. Both the overall biases and the fluctuations in the pseudorange error are simulated consistent with the real measurement in these periods. However, during the period around 850–3200 epoch, the simulator cannot find another valid signal other than the dominating reflecting signal from ray-tracing. Therefore, only the enormous delay in the pseudorange is simulated, but the large fluctuation is missed. Despite this, the proposed simulator can still mimic the dominating pseudorange error due to reflection, which has the most impact on the positioning performance in the urban area.

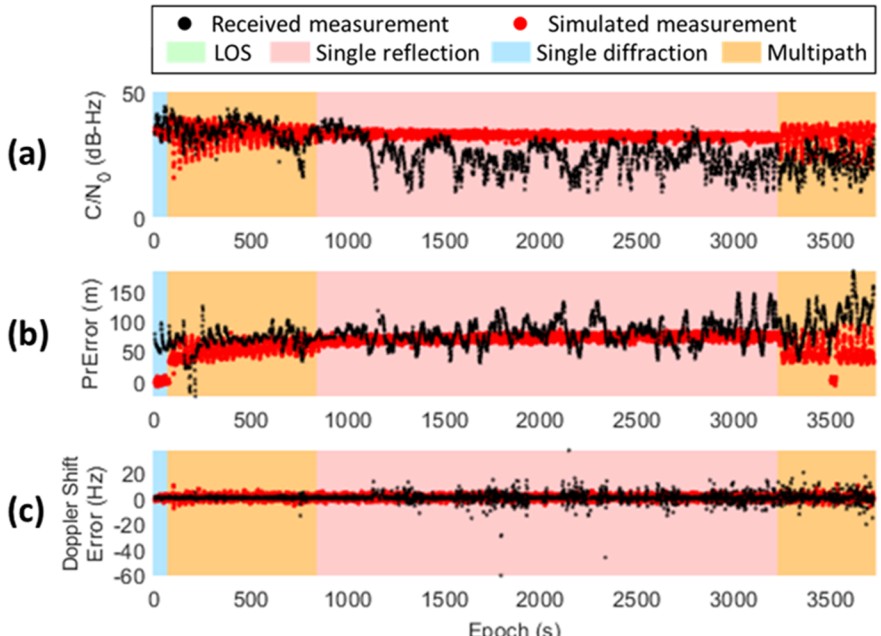

**Figure 13.** The comparison of the simulated and actual received GNSS measurements from G29 satellite during the long-period static experiment, including: (**a**) $C/N_0$; (**b**) pseudorange error; (**c**) Doppler shift error. The colored background denotes the measurement status being categorized during the simulation, including: LOS, single reflection, single diffraction, and multipath.

As the measurements have the greatest impact on the positioning performance, the GNSS $C/N_0$ and pseudorange simulation results for other satellites are shown in Figures 14 and 15, respectively. Table 4 shows the difference in the mean value and the standard deviation (in bracket) of the GNSS measurement parameters ($C/N_0$, pseudorange error, and Doppler shift error) between simulation and real collected data. The difference in mean value may indicate the performance on simulating the overall bias behavior, while the difference in standard deviation may indicate the performance on simulating the noise behavior. For most of the satellites, the simulated $C/N_0$ measurements are consistent with the real measurements, especially many of them appropriately simulate the attenuation on signal strength due to reflection or diffraction. On the other hand, the simulated pseudorange error is consistent with the real measurement in most epochs, even for those satellites with enormous delay due to reflection. However, for some of the satellites, the simulated $C/N_0$ and pseudorange are stable and healthy, while the real measurement is biased or noisy with fluctuations. This is probably due to the limitation of the ray-tracing technique, only considering the level-of-detail one (LoD-1) building model. The ray-tracing technique cannot trace those valid reflection or diffraction paths relate to the detailed structure of the building. As a result, the corresponding interferences are missed during

the simulation. Despite this, those missed interferences are relatively small compared to those interferences introduced by severe reflections or diffractions, which are properly simulated. Moreover, the real measurement has a more stochastic behavior involving complicated noises and interferences, while the simulated measurement is dominated by a model-based deterministic effect that is more repeatable during the test. In general, the proposed simulator is able to simulate the measurement behavior of each satellite in the urban area in a realistic manner.

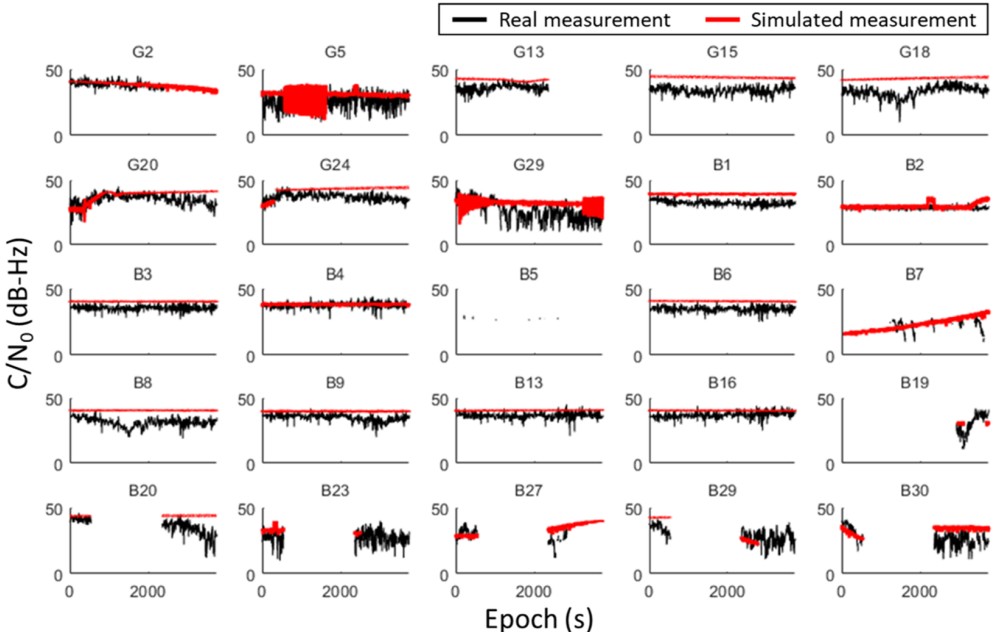

**Figure 14.** The simulated $C/N_0$ and corresponding real measurements from each satellite during the long-period static experiment. The notation below each subfigure denotes the index of each satellite from different constellation, where 'G' —GPS; 'B'—Beidou.

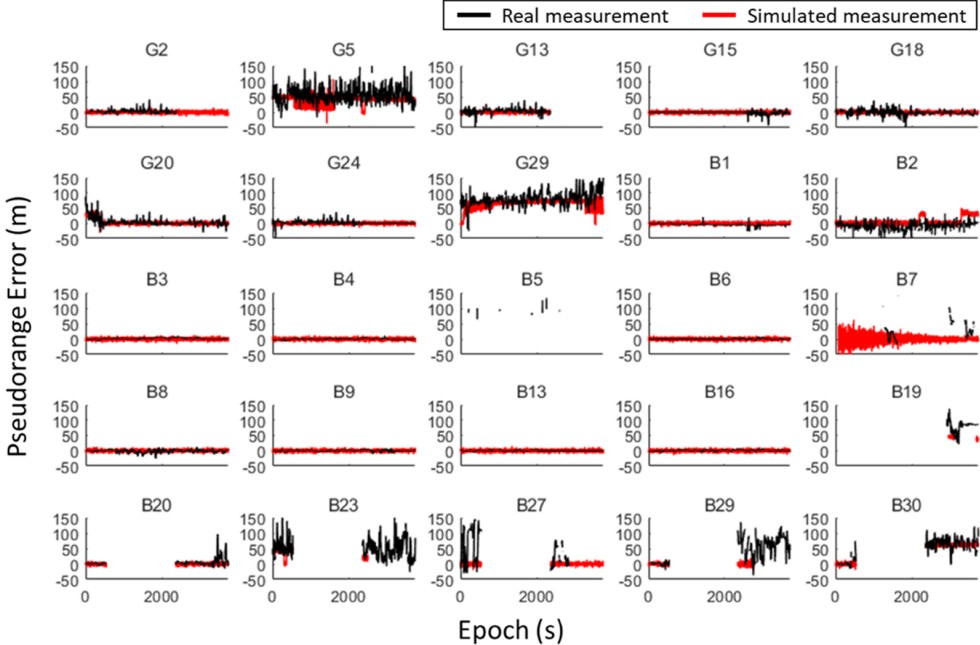

**Figure 15.** The simulated pseudorange error and corresponding labeled value in real measurements for each satellite during the long-period static experiment.

**Table 4.** The difference in mean value (standard deviation) between simulation and real data.

| PRN | Difference on Mean (Standard Deviation) Value | | | PRN | Difference on Mean (Standard Deviation) Value | | |
|---|---|---|---|---|---|---|---|
| | $C/N_0$(dB-Hz) | Pseudorange Error (meter) | Doppler Shift Error (Hz) | | $C/N_0$(dB-Hz) | Pseudorange Error (meter) | Doppler Shift Error (Hz) |
| G02 | 0.6 (0.0) | 2.5 (1.4) | 0.0 (8.2) | B06 | 5.8 (1.4) | 0.5 (1.9) | 0.1 (0.7) |
| G05 | 4.9 (1.2) | 20.1 (8.5) | 0.1 (1.0) | B07 | 0.8 (0.8) | 27.2 (16.4) | 0.1 (1.1) |
| G13 | 5.9 (1.8) | 4.5 (3.5) | 0.2 (0.2) | B08 | 9.0 (2.9) | 0.9 (0.9) | 0.1 (0.3) |
| G15 | 9.8 (2.1) | 2.6 (3.3) | 0.3 (0.4) | B09 | 4.8 (2.0) | 0.3 (1.6) | 0.0 (0.7) |
| G18 | 9.9 (3.4) | 0.3 (4.0) | 0.2 (0.4) | B13 | 3.8 (1.5) | 0.3 (1.9) | 0.0 (0.7) |
| G20 | 3.3 (0.4) | 1.5 (3.2) | 0.2 (0.2) | B16 | 3.2 (1.6) | 0.9 (1.7) | 0.1 (0.7) |
| G24 | 5.5 (0.6) | 2.4 (2.6) | 0.3 (0.1) | B19 | 0.0 (6.4) | 37.1 (11.8) | 0.3 (0.8) |
| G29 | 6.3 (4.5) | 16.4 (5.8) | 0.0 (1.3) | B20 | 8.0 (5.6) | 5.5 (8.4) | 0.0 (0.2) |
| B01 | 6.8 (1.3) | 4.7 (1.0) | 0.2 (3.4) | B23 | 5.1 (2.3) | 15.4 (13.1) | 0.0 (1.0) |
| B02 | 2.1 (1.0) | 11.5 (3.7) | 2.2 (38.2) | B27 | 5.8 (0.0) | 45.8 (46.9) | 0.3 (0.1) |
| B03 | 4.8 (1.2) | 1.6 (1.6) | 0.0 (0.5) | B29 | 5.4 (3.1) | 43.4 (32.3) | 0.0 (0.5) |
| B04 | 0.6 (1.0) | 11.5 (3.7) | 2.2 (38.2) | B30 | 5.9 (3.4) | 4.5 (4.7) | 0.1 (0.9) |
| | | | | All | 4.2 (0.5) | 5.9 (8.7) | 0.1 (6.3) |

The least squares positioning results using the simulated measurements are shown in Figure 16, compared with the positioning result using real measurements. Although the positioning result based on the real measurement has a larger variance than the simulation-based solutions, both the positioning solutions have a similar error distribution mainly along the road direction. The corresponding positioning errors during the experiment are shown in Figure 17 with respect to East–West (E-W) and North–South (N-S) direction. The positioning errors from simulation have the same trend with the positioning errors from real measurements, excluding few epochs that the large errors are unable to simulate. This is probably due to the limitation of the ray-tracing and the model accuracy. In summary, the proposed simulator can realistically simulate the GNSS measurement and corresponding positioning error behavior in the urban area during the long-period static experiment.

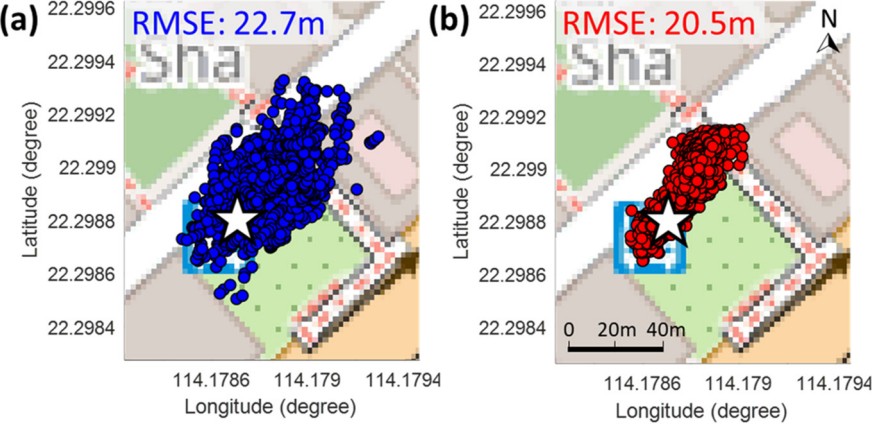

**Figure 16.** The least squares positioning solution based on (**a**) real received GNSS measurement; (**b**) simulated GNSS measurement from the proposed simulator. The corresponding 2D positioning root-mean-square errors (RMSEs) are shown within subfigures.

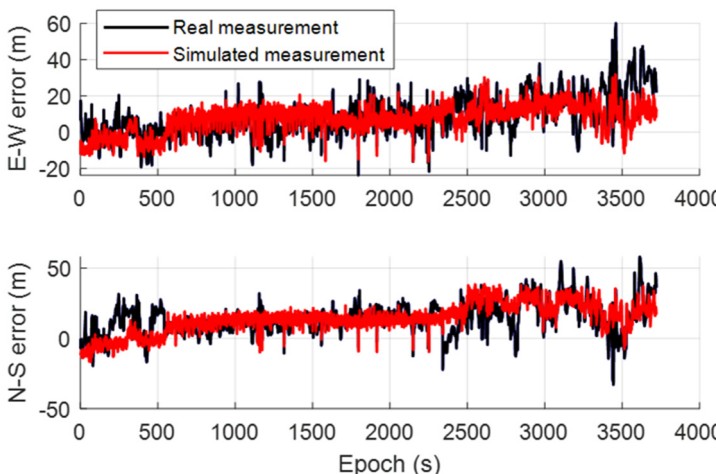

**Figure 17.** The least squares positioning error from the real measurements and simulated measurements during the one-hour static experiment.

### 5.1.3. Dynamic Experimental Verification

During the dynamic experiment, the GNSS measurements from each satellite available in the ephemeris are simulated by the proposed simulator (please see Supplementary Materials). The simulation result of the B1 satellite is shown in Figure 18, compared to the real received measurements. Most of the time, the satellite elevation angle is higher than the building boundary. The proposed simulator appropriately categorizes it as LOS or multipath dominated by the direct signal, and only a small pseudorange error is simulated, which matches the real measurement in the experiment. Besides, during the epoch around 430 and 700, the proposed simulation can appropriately simulate the attenuation of $C/N_0$ due to multipath (interference between two diffracted signals) and reflection. The slight increment of pseudorange error during 660–780 epoch due to reflection from the simulation is also consistent with the real measurements. Notice that the real received $C/N_0$ has additional attenuation compared to the simulation in some epochs. This could possibly be due to the interferences from the signal transporting along an unexpected path that miss-detect by the ray-tracing algorithm. For example, during the first 170 epochs, the vehicle is nearby a tall building still under construction without a proper model. The interferences related to that building cannot be simulated correctly. In general, the simulated GNSS measurement show good consistency with real measurements.

On the other hand, the simulation results of the satellite B6 are demonstrated in Figure 19. The GNSS measurement status is frequently changing between four cases during the simulation since the surrounding environment is rapidly changing during the experiment. Similar to the preceding result, the simulated measurement of $C/N_0$ and pseudorange error from this satellite are consistent with the real measurement, including the signal attenuation/delay due to the multipath effect. For the Doppler shift, the proposed simulator can simulate the error with a level and trend consistent with the real measurement under a rapidly changing environment. Notice that there may still be few epochs that the real measurement has greater degradation than simulation, due to the receptions of unexpected interfered signals. From a simulation point of view, the proposed simulator can instantly determine the measurement status based on the surrounding environment. The measurement behavior for each status is comprehensively simulated with reasonable magnitude and fluctuation. The rapid change of measurement behavior also reflects the unstable measurement status and the surrounding environment during the dynamic operation. It is much more realistic than the statistical noise model that is normally used to validate the GNSS positioning algorithms in the urban area.

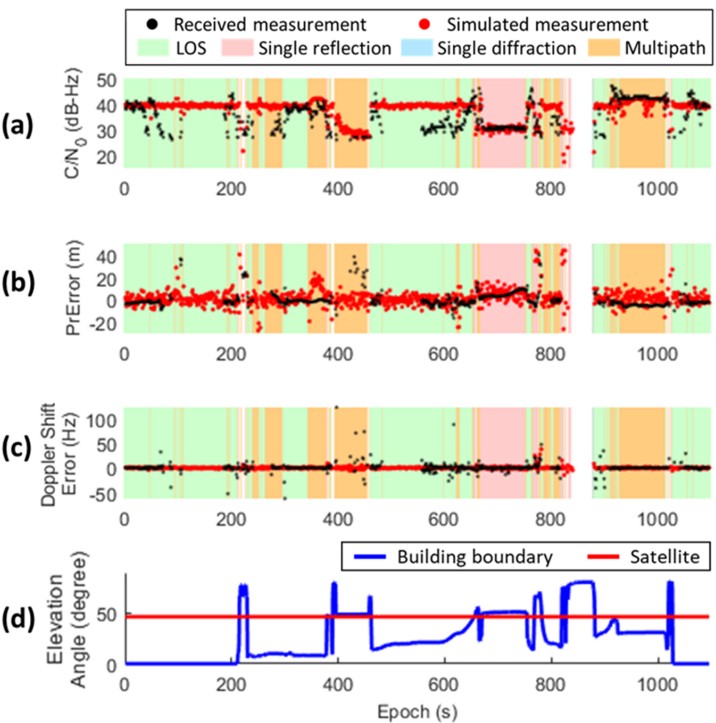

**Figure 18.** The comparison of the simulated and actual received GNSS measurements from B1 satellite during the dynamic vehicular experiment, including: (**a**) $C/N_0$; (**b**) pseudorange error; (**c**) Doppler shift error; (**d**) the elevation angles of the satellite and the building boundary on the satellite's azimuth direction. The colored background denotes the measurement status being categorized during the simulation, including: LOS, single reflection, single diffraction, and multipath.

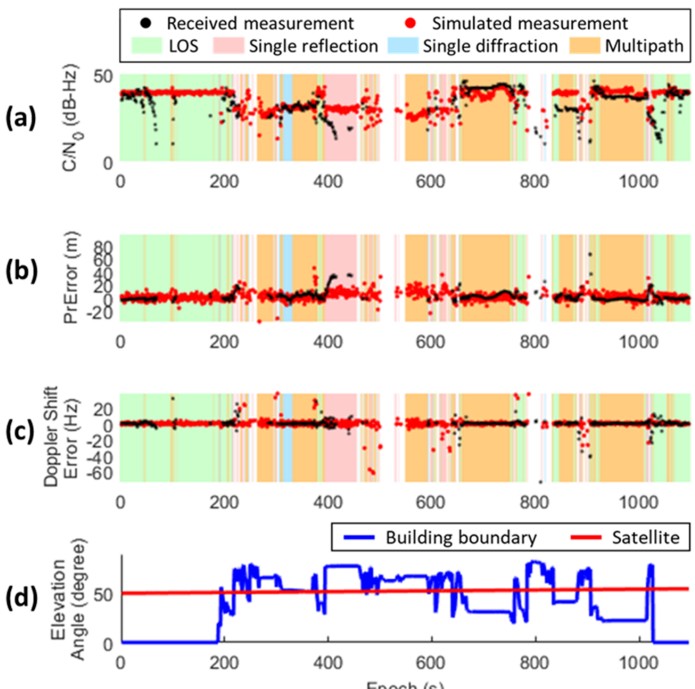

**Figure 19.** The comparison of the simulated and actual received GNSS measurements from B6 satellite during the dynamic vehicular experiment, including: (**a**) $C/N_0$; (**b**) pseudorange error; (**c**) Doppler shift error; (**d**) the elevation angles of the satellite and the building boundary on the satellite's azimuth direction.

The overall positioning solutions and corresponding errors from simulation and real measurements are shown in Figure 20. The proposed simulator appropriately simulates the increment of positioning error when entering a dense urban area, and the recovery of positioning accuracy when back to an open-sky environment. The positioning performance based on simulation behaves similar to that based on real measurements, except the beginning and ending periods. As mentioned before, the vehicle is nearby a constructing building without a model in the beginning period. Hence, the measurement errors from that building are underestimated, leading to a much smaller positioning error from the simulation. The second inconsistent period is probably because the vehicle is nearby the building with a complicated (ship-like) structure. The ray-tracing algorithm based on the LoD-1 building model cannot comprehensively predict all the valid interfered signals related to this building. As a result, the corresponding GNSS measurement error and the positioning error are underestimated. In summary, the proposed simulator can realistically simulate the GNSS measurement and positioning behavior of a vehicular agent in the urban area.

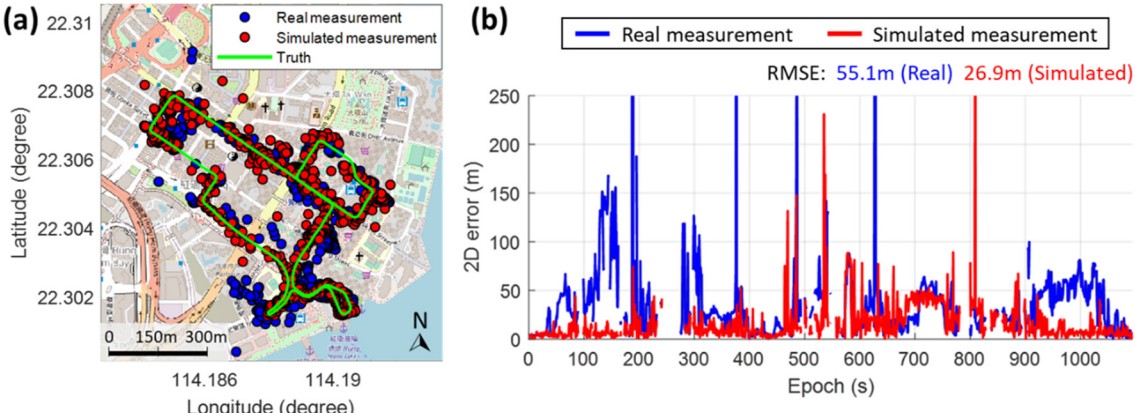

**Figure 20.** (**a**) The positioning solutions from real measurements (blue) and simulated measurements (red) during the dynamic experiment; (**b**) the corresponding least squares positioning 2D errors and the overall RMSE from the simulation and real measurements.

### 5.1.4. Multiagent Performance Verification

During the multiagent experimental verification, six receivers are set up at the designed locations (Figure 10) and collect the GNSS simultaneously. On the other hand, the proposed simulator simulates the GNSS measurements of those receivers during the same period based on the receiver locations, satellite ephemeris, and 3D building model (please see Supplementary Materials). After that, the GNSS positioning behavior from the simulator is compared with that from real collected measurements to verify the performance of the proposed simulator.

The GNSS least squares positioning results of each agent based on real measurements and simulator are shown in Figure 21. The corresponding positioning 2D errors are shown in Figure 22 and summarized in terms of RMSE in Table 5. For Agent 1 and Agent 2 located in the open-sky scenario, the proposed simulator classifies most of the measurements as healthy. Only small errors are simulated in the measurements, which provides accurate positioning solutions consistent with the real measurement performance. Note that the positioning result from simulation contains higher noise than the real measurements. This could possibly be because of the measurement filtering technique that is usually embedded in the commercial GNSS receiver. For the agents other than Agents 1 and 2, the positioning error distribution from the proposed simulation is very similar to that from the real measurement. Especially for Agent 4, the increment of positioning error is also simulated at the epoch consistent with the real measurement. In summary, the proposed

simulator can simultaneously simulate the GNSS measurements of multiple agents with realistic error behaviors in the urban area.

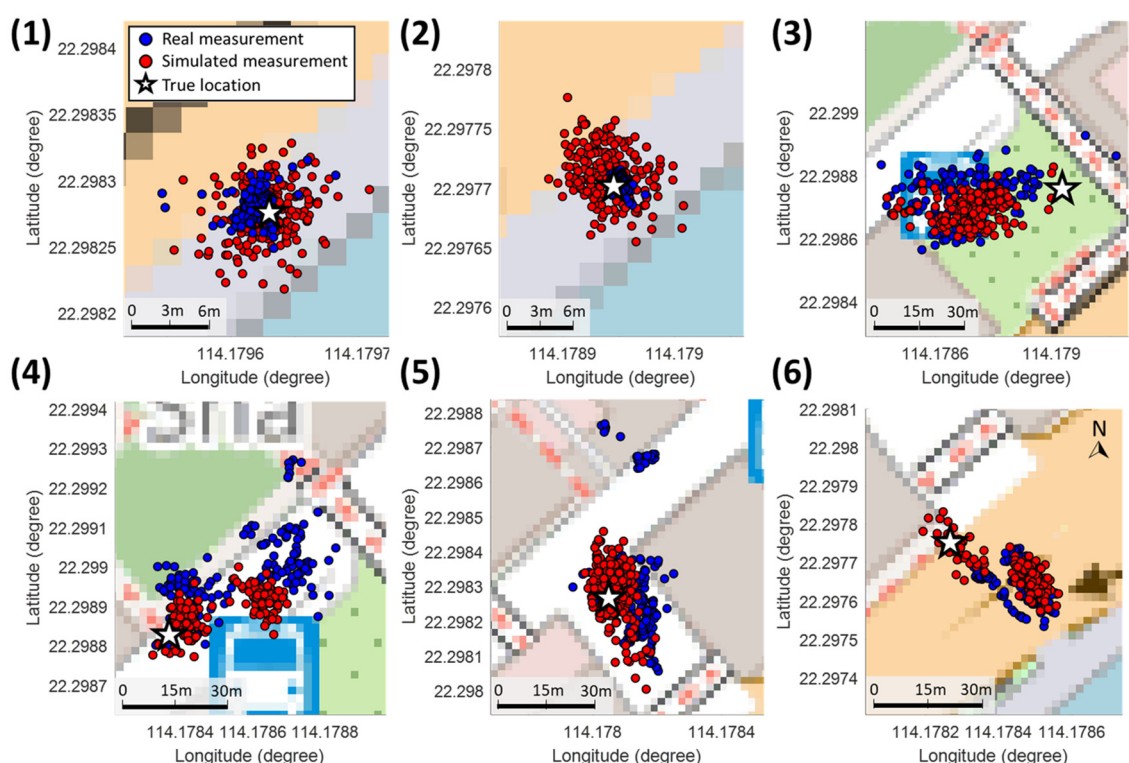

**Figure 21.** The positioning solutions of different agents based on real measurements (blue) and simulated measurements (red) during the multiagent experiment. The number denotes the agent index corresponding to Figure 10.

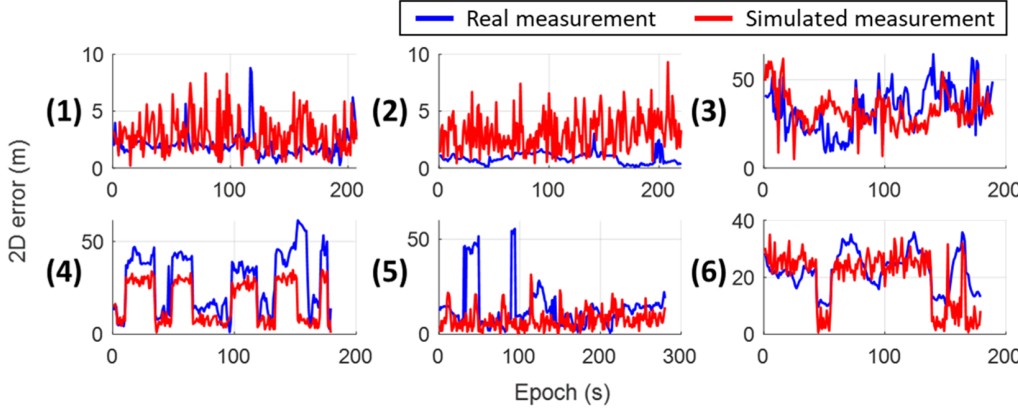

**Figure 22.** The least squares positioning 2D errors of different agents based on the simulated and real measurements during the multiagent experiment. The number is corresponding to the agent index in Figure 21.

**Table 5.** The 2D positioning RMSE (in meter) from the simulated and real measurements of different agents during the multiagent experiment.

|  | Agent | 1 | 2 | 3 | 4 | 5 | 6 |
|---|---|---|---|---|---|---|---|
| RMSE | Real measurement | 2.2 | 1.0 | 35.4 | 32.4 | 17.9 | 23.3 |
|  | Simulated measurement | 3.3 | 3.5 | 34.2 | 20.8 | 9.1 | 22.0 |

*5.2. Simulation Performance for Positioning*

5.2.1. Simulation Setup

Besides validating the consistency between the proposed simulator and the real measurements, it is also important to evaluate the positioning performance after applying different algorithms. Many algorithms are developed based on a certain assumption; for example, the measurement error follows the normal distribution. If only the naive errors are simulated in the measurements, these errors can be easily mitigated by applying advanced positioning algorithms, resulting in an unrealistic outstanding performance. A realistic simulator needs to generate the measurements that maintain the difficulties in urban GNSS positioning, which can be employed to appropriately evaluate or improve the performance of various positioning algorithms.

To evaluate the positioning performance based on the measurements from the proposed simulator, a realistic multiagent positioning scenario in the urban area is generated from SUMO [36] beforehand. The trajectory of each agent from SUMO considering urban transportation behaviors is shown in Figure 23. Five road agents are simulated, including one pedestrian in the open-sky area (Agent 1), one pedestrian in the dense urban area (Agent 2), and three vehicular agents operating with changing environments (Agents 3–5). After simulating the GNSS measurements corresponding to each agent trajectory by the proposed simulator, the simulated measurements are applied with six different urban GNSS positioning algorithms as follows:

(1)    LS: conventional least squares positioning for a single agent;
(2)    CC: least squares positioning with consistency check for a single agent [60];
(3)    RT: 3DMA GNSS ray-tracing positioning for a single agent [23];
(4)    RT-CP: 3DMA GNSS ray-tracing based collaborative positioning with factor graph optimization [26].

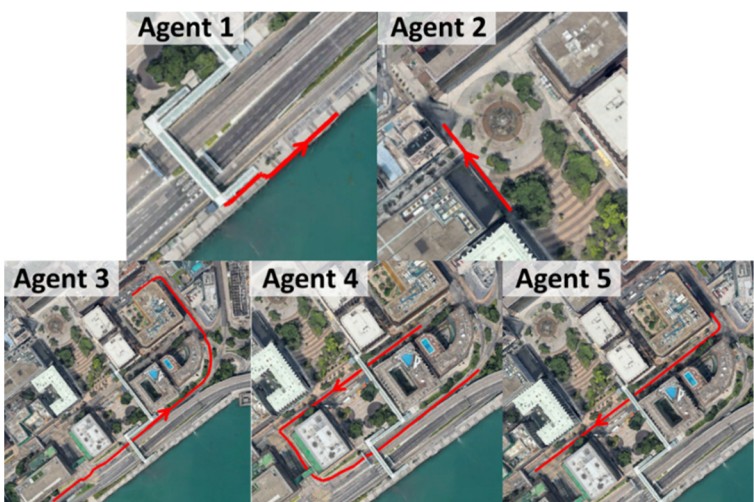

**Figure 23.** The trajectory of each agent simulated from SUMO [36] for the proposed multiagent GNSS measurement simulation and positioning performance evaluation. Agents 1 and 2 are simulated as pedestrians while Agents 3–5 are simulated as vehicles. The arrow on the trajectory denotes the moving direction.

5.2.2. Positioning Performance

The solutions of each agent by applying the above four different GNSS positioning algorithms to the simulated measurements are demonstrated in Figure 24. The corresponding 2D positioning error distributions and RMSEs of each agent are shown in Figure 25 and Table 6, respectively. For Agent 1 in the open-sky area, all four positioning algorithms have similar accurate performances, since most of the measurements are simulated without interferences from buildings.

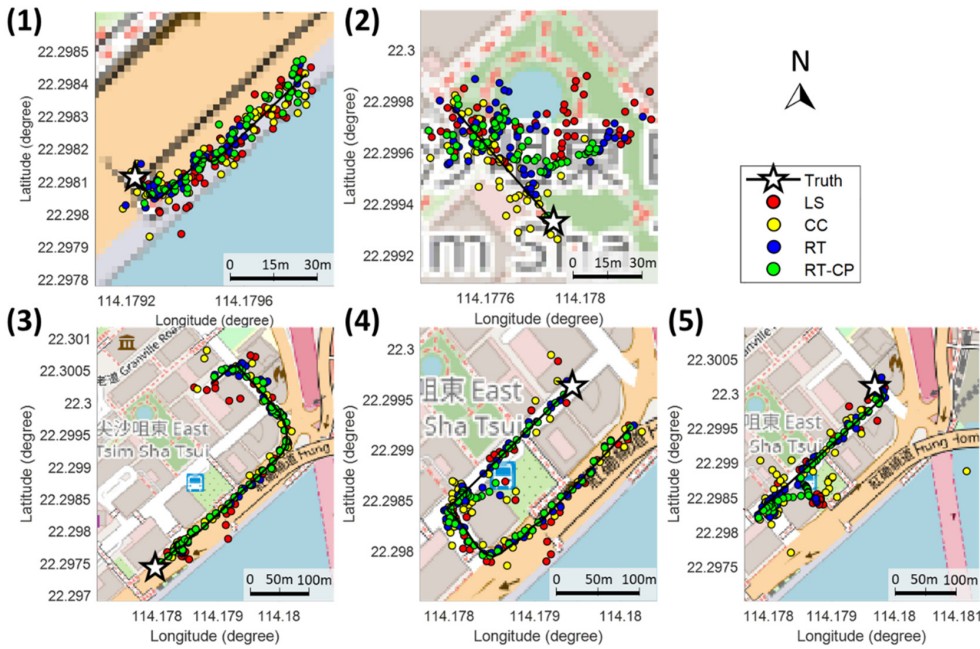

**Figure 24.** The solutions of different positioning algorithms applied on the GNSS measurements from the proposed simulator on different scenarios, including the least squares positioning method (LS), the least squares method with consistency check (CC), the 3DMA GNSS ray-tracing positioning method (RT), and the 3DMA GNSS ray-tracing based collaborative positioning with factor graph optimization (RT-CP). The sub-figures (**1**)–(**5**) are corresponding to the positioning solutions for Agent 1–5 in Figure 23.

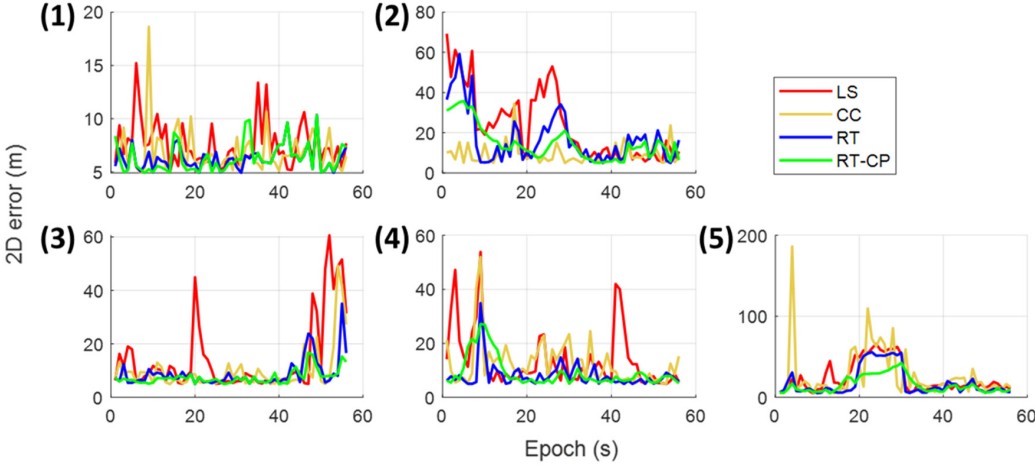

**Figure 25.** The 2D positioning error of different algorithms corresponding to each agent during the multiagent simulation performance verification. The numbers (**1**)–(**5**) are corresponding to Agent 1–5 in Figure 23.

**Table 6.** The root-mean-square error (in meter) of different GNSS positioning algorithms applied on the simulated measurements.

| Agent | 1 | 2 | 3 | 4 | 5 | Expected Performance in a Dense Urban Area |
|---|---|---|---|---|---|---|
| LS | 7.8 | 29.2 | 20.2 | 17.7 | 30.6 | May be over 30 m |
| CC | 7.2 | 10.6 | 12.8 | 15.2 | 41.7 | Around 20 m [60] |
| RT | 7.8 | 20.0 | 10.6 | 9.1 | 24.0 | Around 10–20 m [23] |
| RT-CP | 6.5 | 17.1 | 7.9 | 10.7 | 18.6 | Around 10 m [26] |

For Agent 2 in the urban area, the least-squares positioning RMSE is close to 30 m, which is reasonable for a low-cost GNSS receiver sensitive with the reflection interferences from buildings. In this scenario, the agent moves along one side of the buildings, and most of the reflected signals are probably from the opposite buildings 70 m away. Moreover, the sky-view of Agent 2 is very limited due to building blockage, which makes the multipath effect containing the direct signal less likely to occur. Hence, besides the healthy LOS measurements, the rest are more likely to be the reflected measurements with enormous delay. In this case, those degraded measurements are very inconsistent with other measurements. Therefore, the consistency check method can easily detect and isolate those outliers, achieving excellent positioning performance. On the other hand, the occurrence of multipath effect with two reflected signals is possible for this scenario, which reaches the limitation of the current ray-tracing positioning technique only considers one dominating signal. As a result, not much improvement is achieved by applying the ray-tracing positioning algorithm. Similarly, the ray-tracing based collaborative positioning only slightly improves the positioning accuracy by the aid from neighboring agents.

For vehicular Agent 3 and Agent 4, the surrounding environment is constantly changing, which has a higher chance of experiencing the scenario with complicated measurement error behavior, for example, the intersection with complicated multipath effects or the narrow street with very limited measurement number. Therefore, the consistency check only achieves limited improvements. On the other hand, the ray-tracing algorithm corrects the reflected measurement, which guarantees a sufficient amount of reliable measurement for better positioning accuracy. The ray-tracing based collaborative positioning method further employs the measurements from neighboring agents to eliminate the systematic errors, resulting in a slightly better performance.

For Agent 5 that is always in a narrow street, many of the measurements contain enormous errors, which makes both the LS method and CC method significantly degraded. The ray-tracing method is only capable of correcting the single reflection delay, which limits the positioning improvement. Its extension with collaborative positioning (RT-CP) can only reduce the RMSE to 18.6 m.

In summary, the performances of different positioning algorithms applied to our simulated measurements are reasonable and consistent with the real performance of the low-cost GNSS receiver in the urban area. Therefore, the proposed simulator is capable of providing realistic GNSS measurements of multiple agents in the urban scenario for the study and analysis of collaborative positioning algorithms.

## 6. Discussion

The results from the proposed simulator show great consistency with the real GNSS measurement in the urban area, which validates its capability to provide realistic GNSS measurements for various urban positioning algorithm evaluations or developments. The proposed simulator is developed based on sophisticated models, covering most of the interferences in the urban area. As the results in Table 6, by applying different advanced positioning algorithms, the remaining positioning errors are consistent with the real experimental performances reported in [23,26,60]. Therefore, the proposed simulator can appropriately reflect the challenges of urban GNSS positioning for future studies, especially the potential algorithms that it is hard to conduct experimental verification on, such as the large-scale collaborative positioning algorithms.

Compared to other existing algorithms, the proposed simulator is much less complicated, but still maintains a sufficient verisimilitude level. Besides the measurement availability prediction achieved by the simulator from UCL [61], our simulation further simulates all the basic GNSS raw measurements employed for positioning. Both the conventional GNSS raw measurement simulators, MUSTARD [29] and SNACS [30], simulate the interferences most likely from the ground, whereas our simulator focuses on a more complex urban scenario considering severe interferences coming from buildings. Instead of constructing a virtual environment, our simulator directly uses the 3D building model to

simulate the GNSS measurements reflecting the error behavior on a specific site. Another advanced GNSS simulator, SimGEN$^@$+SE-NAV [31], also employs the ray-tracing algorithm and sophisticated models for the interferences simulation in the urban area. However, it involves complicated radio channel modeling, which may not be useful for verifying the algorithms only using measurement-level data. Moreover, the detailed procedures of interference modeling in [31] are not explained. A recent study employs a reference RF data to simplify the channel modeling during GNSS measurement simulation, but still requires SDR for simulation. Different from the above approaches, our proposed simulator avoids the complicated channel modeling by using an open-sky $C/N_0$ model for reference, and directly simulates measurement-level GNSS data, which is convenient and adequate to verify various positioning algorithms. Each step of the GNSS measurement simulation is introduced in detail, providing a clear and complete guideline for developing a GNSS simulator. Moreover, the modeling of different interferences, including reflection, diffraction, and multipath, is explained comprehensively, which can be extracted for individual study or evaluation by other scholars. The simulator in the current study is developed for the GNSS constellation of GPS and BDS. Its support to GLONASS and GALILEO measurements will be developed with a similar approach in the future.

Besides providing realistic GNSS data, the proposed simulator can, in turn, improve the existing 3DMA GNSS positioning algorithms. Most of the 3DMA GNSS positioning algorithms determine the agent position by searching for a candidate location with the predicted measurement that best matches the real measurement. However, the current methods only conduct matching on the satellite visibility or the direct/reflected pseudo-range [22,23]. The diffraction and multipath models in this study can be used to extend the 3DMA GNSS positioning algorithm by considering the matching on the $C/N_0$ involving diffraction or multipath. The 3DMA GNSS considering the diffraction will be our future work.

However, the proposed simulator still has three limitations. Firstly, the measurement simulation is highly relying on the ray-tracing technique, which is still unable to consider all the interferences from buildings comprehensively. An advanced ray-tracing technique needs to be developed with the consideration of detailed building models and material effects in the future. Moreover, the proposed simulator only provides the measurements based on individual epoch, which neglects the GNSS receiver dynamics during the operation. Therefore, different filtering techniques need to be considered to model the measurement dynamics over time in the future. Finally, the current simulator always produces continuous measurements, even when the satellite may only be barely tracked by the real receiver. The modeling of the signal loss-of-lock and acquisition behavior and the latency caused by the filters in the receiver tracking loops are worth investigating for the proposed simulator in the future to make the measurements more realistic. By involving the tracking-level models in the future, the simulator can be extended to support the realistic modeling of the carrier phase measurement with cycle slip effects in an urban area.

## 7. Conclusions

In this study, a realistic GNSS measurement simulator for multiple agents in the urban area is developed, considering the measurement degradation due to surrounding buildings. Four types of GNSS measurements, including LOS, reflection, diffraction, and multipath, are considered during simulation, covering most of the interferences in the urban area. From each measurement point of view, both the one-hour static and vehicular dynamic experiments validate that the measurement from the proposed simulator has a consistent and reasonable error behavior compared to the real collected measurement. From the overall performance point of view, the multiagent experiment validates that the positioning results based on the measurement from the proposed simulator can realistically reflect the GNSS positioning error behavior in the urban area. Moreover, the proposed simulator is integrated with SUMO to simulate the GNSS measurement of multiple agents considering the transportation behaviors. The simulated measurements are further applied

with advanced positioning algorithms, which verifies that the proposed simulator can appropriately express the current difficulties of precise positioning as well as the bottleneck of different positioning algorithms. Therefore, the proposed simulator can provide realistic GNSS measurements of multiple agents to study and improve the state-of-art GNSS collaborative positioning algorithms in the urban area.

**Supplementary Materials:** This paper provides three examples of the GNSS observation data in the format of the RINEX 3.02. The ground truth of the data is also included. Please note that the GNSS ephemeris data (RINEX nav file) is not included since the data can be easily downloaded online. Please follow the Readme.txt for the detail information on downloading the ephemeris data via HK SatRef (https://www.geodetic.gov.hk/en/satref/satref.htm).

**Author Contributions:** Conceptualization, G.Z. and L.-T.H.; methodology, G.Z. and B.X.; software, G.Z. and H.-F.N.; formal analysis, G.Z., B.X., H.-F.N.; data collection, G.Z., B.X. and H.-F.N.; writing—original draft preparation, G.Z.; writing—review and editing, G.Z., B.X., H.-F.N. and L.-T.H.; supervision, L.-T.H. All authors have read and agreed to the published version of the manuscript.

**Funding:** This work is supported by the Emerging Frontier Areas (EFA) scheme by Research Institute for Sustainable Urban Development, The Hong Kong Polytechnic University—BBWK "Resilient Urban PNT Infrastructure to Support Safety of UAV Remote Sensing in Urban Region".

**Data Availability Statement:** The GNSS simulation data used for verification can be accessed at https://www.polyu-ipn-lab.com/collaborative-positioning-for-iot-i.

**Conflicts of Interest:** The authors declare no conflict of interest.

## Appendix A. Multipath Signal Strength Estimation

To simulate the $C/N_0$ measurement under the joint effect of two signals, the strength of each signal and the interference between two signals are required to be estimated. This interference is also related to the phase difference between two signals. A natural approach to evaluate such joint effect is through the superposition of those two electric fields $\mathbf{E}_a$ and $\mathbf{E}_b$, as follows.

$$\mathbf{E}_{joint} = \mathbf{E}_a + \mathbf{E}_b \tag{A1}$$

$$\mathbf{E}_{a,b} = \begin{cases} \mathbf{E}_{ref} \cdot A\left(s_{ref}\right) \cdot e^{-jks_{ref}}, & a,b \in LOS \\ \mathbf{E}_{D1} \cdot A(s_D) \cdot e^{-jks_D}, & a,b \in diffraction \\ \mathbf{E}_{R1} \cdot A(s_R) \cdot e^{-jks_R}, & a,b \in reflection \end{cases} \tag{A2}$$

where $\mathbf{E}_a$ and $\mathbf{E}_b$ represent the electric field of a direct LOS signal, reflected signal, or diffracted signal. $\mathbf{E}_{ref}$ represents the reference electric field on a specific location along the direct propagation path with a distance $s_{ref}$ to the receiver location. $\mathbf{E}_{D1} = \mathbf{E}_{D0} \cdot \mathscr{D}_{RR}$ represents the diffracted field on the point of diffraction, which can be written as the multiplication of the incident field $\mathbf{E}_{D0}$ on the point of diffraction and the diffraction coefficient $\mathscr{D}_{RR}$. $s_D$ is the transmission distance between the point of diffraction and the receiver. Similarly, $\mathbf{E}_{R1} = \mathbf{E}_{R0} \cdot \mathscr{R}_{LR}$ represents the reflected electric field on the point of reflection, which can be written as the multiplication of the incident field $\mathbf{E}_{R0}$ on the point of reflection and the reflection coefficient $\mathscr{R}_{LR}$. $s_R$ is the transmission distance between the point of reflection and the receiver. $A(s)$ denotes the spreading factor of the electric field related to the propagation distance. The term $e^{-jks}$ accounts for the phase shift during the propagation. For convenience, the preceding expression can be written concerning the same reference field, the electric field on the road agent location.

According to the modern geometrical optics (GO) theory [39], the LOS electric field on the road agent location during the direct propagation with a spherical wavefront can be expressed as

$$\mathbf{E}_{LOS} = \mathbf{E}_{ref} \cdot A\left(s_{ref}\right) \cdot e^{-jks_{ref}} = \mathbf{E}_{ref} \cdot \sqrt{\frac{s'_{ref} \cdot s'_{ref}}{\left(s'_{ref} + s_{ref}\right)\left(s'_{ref} + s_{ref}\right)}} \cdot e^{-jks_{ref}} = \mathbf{E}_{agent} \quad \text{(A3)}$$

where $s'_{ref}$ denotes the distance between the satellite and the reference field. $A\left(s_{ref}\right) = \sqrt{s'_{ref} \cdot s'_{ref} / \left(s'_{ref} + s_{ref}\right)\left(s'_{ref} + s_{ref}\right)}$ is the spreading factor of a spherical wave during free propagation. Then, the electric field of a diffracted signal during the transmission can be expressed via the reference field on the road agent location as follows based on UTD and Equation (A3).

$$\begin{aligned}
\mathbf{E}_{diffraction} &= \mathbf{E}_{D0} \cdot \mathscr{D}_{RR} \cdot A(s_D) \cdot e^{-jks_D} \\
&= \mathbf{E}_{D0} \cdot \mathscr{D}_{RR} \cdot \frac{1}{\sqrt{s_D}} \cdot e^{-jks_D} \\
&= \frac{\mathbf{E}_{agent}}{A\left(s_{LOS} - s'_D\right) \cdot e^{-jk(s_{LOS} - s'_D)}} \cdot \mathscr{D}_{RR} \cdot \frac{1}{\sqrt{s_D}} \cdot e^{-jks_D} \\
&= \frac{\mathbf{E}_{agent}}{\sqrt{\frac{s'_D s'_D}{\left[s'_D + \left(s_{LOS} - s'_D\right)\right]\left[s'_D + \left(s_{LOS} - s'_D\right)\right]}} \cdot e^{-jk(s_{LOS} - s'_D)}} \cdot \mathscr{D}_{RR} \cdot \frac{1}{\sqrt{s_D}} \cdot e^{-jks_D} \\
&\approx \mathbf{E}_{agent} \cdot \frac{\mathscr{D}_{RR}}{\sqrt{s_D}} \cdot e^{-jk(s'_D + s_D - s_{LOS})} = \mathbf{E}_{agent} \cdot \frac{\mathscr{D}_{RR}}{\sqrt{s_D}} \cdot e^{-jk\epsilon_D}
\end{aligned} \quad \text{(A4)}$$

As Figure 2 shows, $s'_D$ is the distance between the satellite and the point of diffraction. The electric field at the point of diffraction and at the road agent (receiver) location has the relationship $\mathbf{E}_{agent} = \mathbf{E}_{D0} \cdot A(s_{LOS} - s'_D) \cdot e^{-jk(s_{LOS} - s'_D)}$. $A(s_{LOS} - s'_D) = \sqrt{s'_D s'_D / \left[s'_D + \left(s_{LOS} - s'_D\right)\right]\left[s'_D + \left(s_{LOS} - s'_D\right)\right]}$ is the spreading factor from the field equivalent to the point of diffraction to the receiver under free propagation. $A(s_D) = 1/\sqrt{s_D}$ is the spreading factor of the diffracted field to the receiver. Since the satellite is much further away compared to the referencing point for the receiver, $A(s_{LOS} - s'_D)$ is reduced to unity. Similarly, the reflected electric field on the road agent location can be written as follows.

$$\begin{aligned}
\mathbf{E}_{reflection} &= \mathbf{E}_{R0} \cdot \mathscr{R}_{LR} \cdot A(s_R) \cdot e^{-jks_R} \\
&= \mathbf{E}_{R0} \cdot \mathscr{R}_{LR} \cdot \sqrt{\frac{\wp_{R1} \wp_{R2}}{(\wp_{R1} + s_R)(\wp_{R2} + s_R)}} \cdot e^{-jks_R} \\
&= \frac{\mathbf{E}_{agent}}{A\left(s_{LOS} - s'_R\right) \cdot e^{-jk(s_{LOS} - s'_R)}} \cdot \mathscr{R}_{LR} \cdot \sqrt{\frac{\wp_{R1} \wp_{R2}}{(\wp_{R1} + s_R)(\wp_{R2} + s_R)}} \cdot e^{-jks_R} \\
&= \frac{\mathbf{E}_{agent}}{\sqrt{\frac{s'_R s'_R}{\left[s'_R + \left(s_{LOS} - s'_R\right)\right]\left[s'_R + \left(s_{LOS} - s'_R\right)\right]}} \cdot e^{-jk(s_{LOS} - s'_R)}} \cdot \mathscr{R}_{LR} \cdot \sqrt{\frac{\wp_{R1} \wp_{R2}}{(\wp_{R1} + s_R)(\wp_{R2} + s_R)}} \cdot e^{-jks_R} \\
&\approx \mathbf{E}_{agent} \cdot \mathscr{R}_{LR} \cdot e^{-jk(s'_R + s_R - s_{LOS})} = \mathbf{E}_{agent} \cdot \mathscr{R}_{LR} \cdot e^{-jk\epsilon_R}
\end{aligned} \quad \text{(A5)}$$

where $A(s_R) = \sqrt{\wp_{R1} \wp_{R2} / (\wp_{R1} + s_R)(\wp_{R2} + s_R)}$ is the spreading factor of the reflected field to the receiver. $\wp_{R1}$ and $\wp_{R2}$ denote the principal radii of the reflected wavefront at the point of reflection. Since the satellite is much further away from the receiver than the referencing point and the point of reflection, both $A(s_{LOS} - s'_R)$ and $A(s_R)$ are reduced to unity, as the near-zone observation [39].

Therefore, the strength ratio between the multipath joint field and the unobstructed field on the road agent location can be expressed by

$$\Gamma_{multipath} = \frac{\mathbf{E}_{joint}}{\mathbf{E}_{agent}} = \Gamma_a + \Gamma_b \tag{A6}$$

where $\Gamma_a$ and $\Gamma_b$ represent the stand-alone strength ratio with respect to the unobstructed field for different cases from Equation (23).

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
