# Peer review of "GNSS RUMS: GNSS Realistic Urban Multiagent Simulator for Collaborative Positioning Research"

_remotesensing, doi:10.3390/rs13040544_

Round 1
Reviewer 1 Report
The publication ‘GNSS RUMS: GNSS Realistic Urban Multi-agent Simulator for Collaborative Positioning Research ' is interesting and well prepared. Congratulations to the authors as they introduced multi-disciplinary knowledge to the created algorithm. The paper is written in professional English. A few editorial corrections are needed. The research is well motivated and supported by previous research. Referenced to other studies are carefully applied, only in places, some citations are missing. The research undoubtedly provides an advance in the current level of the satellite geodesy field. Methodology and field tests are described in detail. Discussion is also generally correct, however, some numbers related to the other simulation models would be valuable for the reader. The graphics in the papers are well designed. In some cases, eg. Fig. 6 the a)-b) identifiers could have a smaller font size. The authors did not indicate in the introduction section the number of GNSS systems used. Later in the text (row 294-295), the reader finds that only GPS and BDS are used. It is a bit misleading as also GLO and GAL might be expected to appear in the research.
The paper is almost ready for publication. Several minor changes in the text are advisable:
- Rows 39-40: Multipath description could be omitted as it is a very sophisticated paper for professionals. Please note that the authors did not explain by description e.g. numerous physical factors that undoubtedly are less known to the GNSS-related readers
- Row 262, formula (1) – please explain why the authors introduced the 10xlog10() function to the simulation. Why such parameters were chosen? Similar cases: row 332, eq. (9) and row 359, eq. 14
- Rows 290-292: citation needed for the thesis support
- Fig 3. – please explain why the authors analyzed only one SV from each GNSS.Do the results from the whole constellation are similar?
- Row 365: typo (capital letter)
- Rows 401: please explain better why the authors used the 20dB threshold value. Citation needed for support
- Row 422: editorial error (indentation)
- Row 429, fig. 4: editorial error (bold)
- Rows 595-596: please explain why the authors used G Earth coordinates instead of GNSS RTK/RTN measurement as a reference value. RTK/RTN would be a more reliable source.
- Row 623: “consistent” is too general expression. Some numbers, percentage changes would be valuable for illustrating the results of the simulation
- Row 632, fig. 11: a)-panel vertical axis’ value range is misleading. 30-50 would be more illustrative. A similar case in Fig. 12 c) – one reading from ~2400 should be omitted
- 723, fig. 16: some information on the accuracy obtained would be valuable. The field literature contains publications presenting various types of stationary and dynamic accuracy estimators, eg. 10.1371/journal.pone.0219890, 10.1371/journal.pone.0215562, 10.1088/1361-6501/ab75b2. A similar case in fig. 21 measurements.
- Rows 907-953, some number describing the improvement of the simulation models performance would be very valuable. Eg. using the data from the research of the MUSTARD model, calculating some parameters, and comparing them. This is a soft spot of the research.
Author Response
Thank you for your time and effort on our paper. The point-to-point rebuttal letter is attached.

Reviewer 2 Report
Review is in the attached PDF file.

Author Response

(The authors gave the same response as above.)

Reviewer 3 Report
The authors describe and praise their GNSS measurement simulator for urban environments. The first part of the paper presents a complete overview of the error terms considered, and the second part presents a comparison of simulated and real-life results. The paper is generally well written. My main remark is about the overly optimistic interpretation of the results. For all tests, the authors claim a good match between their simulation and real data, but this is not sufficiently based on facts and many presented tests results seem to prove otherwise in my opinion.
For examples, the measurement results in Fig 14, 15 or the position results in Fig 16, 20 and 22 are difficult to interpret. I wouldn't say that they demonstrate a good match, but this is subject to defining what a "good match" means. To remain factual, I would suggest adding some quantitative statistics (e.g. histogram of errors, standard deviation, min/max…) rather than only providing qualitative statements that are always subject to interpretation.
In Lines 810-812, it is said that, for other receivers, the positioning results match better. This is possible, but this needs to be shown in the paper. Why have the authors decided to publish the results of a comparison with a receiver that exhibits poor match if better results exist?
Likewise, line 914 claim that the results of table 4 are consistent with real experiments reported in other papers. This needs to be substantiated: those results from other papers should be put in table 4 for easier comparison.
In my opinion, an important aspect that is missing in the simulator is the loss-of-lock and acquisition behavior of the receiver. The effect is clearly visible in fig 14&15, where the simulator produces continuous data for some satellites that are barely tracked by the receiver. It would be fair to add a remark on this after line 953 when discussing the limitations and future work.
Other remarks:
- There is a problem in the reference to equations: please review and correct all equation numbering in the text. Most of the references to equations are wrongly numbered.
- Explain how the carrier phase has been validated. The results only address pseudorange/CN0/Doppler. The relevance of the carrier phase in this study is not clear, and the simulator seem to make unrealistic assumptions regarding the absence of cycle slips (line 278).
- Equation (3): explain why a different methodology is used for phase and pseudorange: why is the multipath case treated separately for carrier phase and not for pseudorange?
- Line 298: ‘first-order polynomial fitting...”. The figure clearly does not show first-order polynomials. To be explained.
- Line 299: Hz should be dB-Hz.
- Figure 3: explain how the green dots have been obtained (which receiver, which antenna, which environment).
- Line 324 is a copy of the previous sentence. To be deleted.
- Line 361: replace “r_R” by “s_R”.
- Line 364-366 is copy pasted from another paragraph and should be deleted.
- Eq 26: the envelope should not depend on the phase (beta). Why is this an argument of the multipath envelope function?
- Figure 4: what are the values of d and beta in this figure?
- The manuscript should be proofread carefully, especially with respect to verb conjugation. Examples:
- Line 217: “that obstructing the…”
- Line 243: “the point of reflection fulfills the geometrical relationship is located…”
- Line 265: “is a random variable follows Gaussian distribution…”
- Line 338: “when only considers…”
- Line 465: “is consist of the…”
- Line 514: “is an unknown need to be estimated…”
- Line 577: “is expected to under severe degradation…”
- Line 660: “unfounded relfections”
- Line 719: “that unable to be traced”
- Line 947: "which still unable"
Author Response

(The authors gave the same response as above.)

Round 2
Reviewer 2 Report
I accept the text in present form
Reviewer 3 Report
The authors have adequately addressed all comments, and the manuscript can be accepted.